# Partner independent fusion gene detection by multiplexed CRISPR-Cas9 enrichment and long read nanopore sequencing

Christina Stangl[1,2,3], Sam de Blank [1], Ivo Renkens[1], Liset Westera[4,5], Tamara Verbeek[1], Jose Espejo Valle-Inclan[1,3], Rocio Chamorro González [6,7], Anton G. Henssen [6,7,8,9], Markus J. van Roosmalen[3,4], Ronald W. Stam[4], Emile E. Voest [2,3], Wigard P. Kloosterman [1], Gijs van Haaften [1,10] & Glen R. Monroe [1,10 ✉]

Fusion genes are hallmarks of various cancer types and important determinants for diagnosis, prognosis and treatment. Fusion gene partner choice and breakpoint-position promiscuity restricts diagnostic detection, even for known and recurrent configurations. Here, we develop FUDGE (FUsion Detection from Gene Enrichment) to accurately and impartially identify fusions. FUDGE couples target-selected and strand-specific CRISPR-Cas9 activity for fusion gene driver enrichment — without prior knowledge of fusion partner or breakpoint-location — to long read nanopore sequencing with the bioinformatics pipeline NanoFG. FUDGE has flexible target-loci choices and enables multiplexed enrichment for simultaneous analysis of several genes in multiple samples in one sequencing run. We observe on-average 665 fold breakpoint-site enrichment and identify nucleotide resolution fusion breakpoints within 2 days. The assay identifies cancer cell line and tumor sample fusions irrespective of partner gene or breakpoint-position. FUDGE is a rapid and versatile fusion detection assay for diagnostic pan-cancer fusion detection.

[1] Department of Genetics, Center for Molecular Medicine, University Medical Center Utrecht and Utrecht University, Utrecht, Netherlands. [2] Division of Molecular Oncology, Netherlands Cancer Institute, Plesmanlaan, Amsterdam, Netherlands. [3] Oncode Institute, 3521 AL Utrecht, Netherlands. [4] Princess Máxima Center for Pediatric Oncology, Utrecht, Netherlands. [5] Dutch Childhood Oncology Group (DCOG), Den Haag, Netherlands. [6] Department of Pediatric Oncology/ Hematology, Charité-Universitätsmedizin Berlin, Berlin, Germany. [7] Experimental and Clinical Research Center (ECRC) of the MDC and Charité Berlin, Berlin, Germany. [8] German Cancer Consortium (DKTK), Partner Site Berlin, and German Cancer Research Center (DKFZ), Heidelberg, Germany. [9] Berlin Institute of Health, Berlin, Germany. [10]These authors contributed equally: Gijs van Haaften, Glen R. Monroe. ✉email: g.monroe@umcutrecht.nl

Fusion genes are hallmarks of many human cancers. Recent studies suggest that up to 16% of cancers are driven by a fusion gene[1]. Some cancer types, such as prostate cancer or chronic myeloid leukemia (CML), are characterized by a specific fusion gene (*TMPRSS2-ERG* and *BCR-ABL1,* respectively), whereas other cancer types do not show such clear associations[1,2]. Most fusion genes are highly variable with respect to fusion gene configurations and exact breakpoint-locations. Often, one gene is a recurrent fusion partner (e.g., *KMT2A/MLL*, *ALK*) which exhibits a tissue-specific pattern[3]. However, these genes can fuse to a multitude of partners to obtain their oncogenic potential. One striking example is *KMT2A*, formerly known as MLL, which is a prominent fusion partner in pediatric acute myeloid leukemia (AML) and the predominant fusion partner in acute lymphocytic leukemia (ALL) diagnosed in infants (i.e., children <1 year of age), and has been reported with more than 130 different gene configurations[4,5].

Whereas fusion detection is pathognomonic for some cancer types, it is a determinant of prognosis or treatment choices in other cancer types[6,7]. However, the high levels of variability in fusion gene configurations drastically limits diagnostic detection. Current diagnostic strategies include (break-apart) Fluorescence In Situ Hybridization (FISH) and reverse transcription quantitative polymerase chain reaction (RT-qPCR) assays, depending on the knowledge and breakpoint-variability of the fusion partner[7]. However, these assays are laborious and time-consuming and may not identify the fusion partner. Recently, next generation sequencing (NGS) assays which specifically target recurrent fusion partners have been developed and are currently implemented in clinical practice[8,9]. These assays are highly versatile with respect to partner identification and input material (e.g., suitable for DNA isolated from Formalin-Fixed Paraffin Embedded tissue blocks; FFPE), but are accompanied with longer turnaround-times, increased costs and bioinformatic challenges.

Recent long read sequencing technologies such as Oxford Nanopore Technology (ONT) sequencing have proven immensely helpful in elucidating structural variation in human genomes[10]. Furthermore, the real-time sequencing capabilities yield abundant opportunities for clinical applications. However, sequencing throughput from one nanopore flow cell (2–5x genome coverage; R9.4) is insufficient to elucidate the complete structural variation (SV) landscape of a genome[11]. ONT recently released a Cas9-based protocol for enrichment of specific genomic regions, which utilizes the upstream (5′) and downstream (3′) flanking sequences of the region of interest (ROI), to excise the latter and perform targeted sequencing[12]. Two publications have utilized this method to study methylation and structural variants[12], as well as genome duplications[13]. With this technique, a median on-target coverage of 165x and 254x was achieved, respectively, offering a unique tool to sequence SVs such as fusion genes. However, this approach requires knowledge of both flanking sequences of the ROI, which again restricts its application to detection of only known fusion gene partner combinations.

We here develop FUDGE (FUsion Detection from Gene Enrichment) as a fusion gene identification strategy to perform targeted enrichment of fusion genes and identify — without prior knowledge — the unknown fusion partner and precise breakpoint by using long read, real-time ONT sequencing. Furthermore, we create and implement a complementary bioinformatic tool, NanoFG, to detect fusion genes from long read nanopore sequencing data. Utilizing this approach, we achieve an average breakpoint-spanning coverage of 68x — resulting in an average enrichment of 665x — and identify fusion gene partners from various cancer types (e.g., AML, Ewing Sarcoma, Colon) within 48 h. In addition, we offer strategies for low-input DNA samples

(10 ng), as well as multiplexing of samples and targets to minimize assay costs. Finally, we utilize this method on material in which routine diagnostic procedures were unable to detect the fusion partner, and identify the fusion partner within two days.

## Results

**Schematic overview of fusion gene detection assay.** We developed FUDGE to specifically enrich for fusion genes in which only one gene partner is known and for which the other fusion gene partner and/or breakpoint is unknown. To achieve this, genomic DNA isolated from fresh frozen samples is dephosphorylated as previously described[12] and a crRNA flanking the suspected breakpoint region(s) is utilized to target Cas9 to a specific genomic loci where it creates a double-strand DNA break (Fig. 1a). The Cas9 protein stays predominantly bound to the PAM-distal side of the cut, therefore masking the phosphorylation side on this end, while exposing phosphorylated DNA on the PAM-proximal side of the cut (Fig. 1b). This phosphorylated DNA, following dA-tailing, creates a distinct contact-point that can be used to anneal the ONT-specific sequencing adapters — specifically to this region only. To achieve directionality, the crRNAs are designed in a strand-directed manner to specifically direct reads upstream or downstream of the crRNA sequence — effectively sequencing into the suspected 5′ or 3′ fusion partner (Fig. 1b, Methods, and Supplementary Fig. 1). Thereafter, the enriched libraries are sequenced on one ONT flow cell (R9.4). To robustly detect fusion genes from low coverage nanopore sequencing data, we developed a bioinformatic tool, NanoFG, which reports fusion partners, exact breakpoint-locations, the breakpoint-sequence and primers for validation purposes (Fig. 1c).

**Enrichment and directed sequencing.** To test the ability of the fusion gene detection assay to generate sufficient enrichment and to direct reads in the desired direction, we applied FUDGE to genomic DNA from a male healthy donor. As a proof-of-principle we designed crRNAs for a panel of recurrent fusion partner genes (*BRAF*, *EWSR1*, and *SS18*) in a strand-specific manner. We performed two separate library preparations (PP1 and PP2) and targeted two different exons for each of the three genomic loci per library (Fig. 2a and Supplementary Data 1). As a positive control, we targeted two genomic loci (*C9orf72* and *FMR1*) for which we previously performed targeted sequencing, and used two crRNAs flanking the ROI and with each targeting one of the two different strands (Fig. 2a and Supplementary Data 1). After the sample processing, libraries of PP1 and PP2 were pooled and sequenced on a single flow cell. Sequencing resulted in a throughput of 1.665 Gbs which corresponds to a mean genome coverage of 0.5x (Supplementary Data 1). For the loci where only one strand of the genome was targeted, on average 89% of the reads sequenced in the anticipated 5′ or 3′ direction (Fig. 2b–d and Supplementary Fig. 2a–e). The coverage at the PP1 and PP2 cut-sites were 115x and 102x (*BRAF*) (Fig. 2b), 142x and 101x (*EWSR1*) (Fig. 2c), 117x and 118x (*SS18*) (Fig. 2d), 57x and 104x (*C9orf72*) (Fig. 2e), and 11x and 44x (*FMR1*) (Fig. 2f), respectively. The average read-length was 9.9 kb (Fig. 2g and Supplementary Data 1) and on average 116 reads crossed the most common fusion breakpoint-locations (Fig. 2b–d and Supplementary Data 1), proving the applicability of this assay to detect fusion genes irrespective of breakpoint-position.

**Identification of gene fusions in cancer cell lines.** To test that FUDGE identifies fusion genes independent of targeted gene or breakpoint-location, we applied this technique to three fusion-positive cancer cell lines in which the fusion configuration was

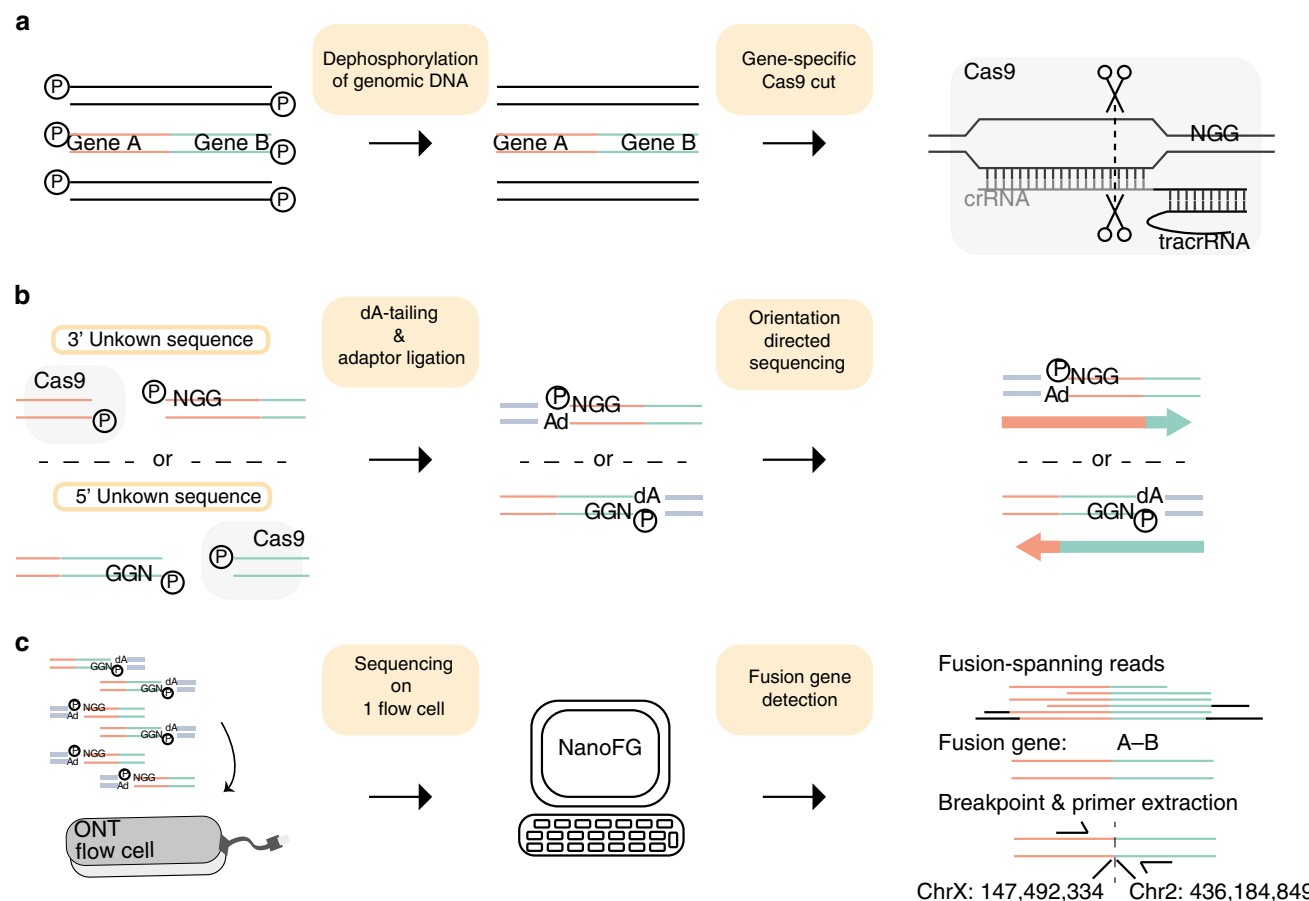

**Fig. 1 Schematic overview of FUDGE. a** Genomic DNA sample is dephosphorylated and crRNA-guided target-specific double-stranded cuts are introduced through Cas9. **b** Phosphorylation-sites are exposed through the double-strand breaks; however, Cas9 remains bound to the PAM-distal side of the cut and blocks phosphorylation of the DNA on this side. DNA-ends are dA-tailed and adapters are ligated only to phosphorylated DNA-ends proximal to the PAM sequence. Sequencing direction is dictated by the adapters, towards the unknown sequence. **c** Targeted libraries are loaded and sequenced on one ONT flow cell (R9.4). NanoFG is run on the nanopore sequencing data, extracts fusion-spanning reads, detects fusion genes and provides exact fusion gene configuration, breakpoint-location, breakpoint-sequence and fusion-spanning primer sequences.

previously identified. The Ewing sarcoma cell lines A4573 (ref. [14]) and CHP-100 (ref. [15]) harbor the *EWSR1-FLI1* fusion gene and the synovial sarcoma HS-SYII cell line contains a *SS18-SSX1* fusion[16]. We targeted three loci per sample (*BRAF* Exon 10, *EWS*R1 Exon 7, *SS18* Exon 9) and sequenced the samples on one flow cell each (Supplementary Data 1). This produced a mean genome coverage of 0.24x (A4573), 0.15x (CHP-100), and 0.015x (HS-SYII) (Fig. 3a). We observed a sharp increase to 81x (A4573), 66x (CHP-100), and 11x (HS-SYII) on-target coverage (cut to breakpoint) due to the achieved directionality (Fig. 3a and Supplementary Fig. 1). This relates to an overall on-target fold-enrichment of 342x (A4573), 443x (CHP-100), and 735x (HS-SYII) (Fig. 3b–e).

To easily identify fusion-spanning reads from nanopore data, we developed NanoFG[17]. NanoFG is an amendment to NanoSV[10] that calls fusion genes from nanopore sequencing data and reports the exact breakpoint-location, breakpoint-sequence and breakpoint-spanning primers for each gene fusion (Fig. 1). The breakpoint was spanned by 69 (A4573), 62 (CHP), and 6 (HS-SYII) reads, which correlates to a 290x, 417x, and 406x enrichment, respectively. NanoFG identified the two *EWSR1-FLI1* fusion genes with 28 (A4573) (Fig. 3a, c) and 18 (CHP-100) (Fig. 3a, d) fusion-spanning reads which relates to a fusion-specific enrichment of 118x and 121x, respectively (Fig. 3b). The two Ewing sarcoma cell lines harbored the same fusion gene, however, with different breakpoint-locations (Supplementary

Fig. 3A). These differences were readily detected by NanoFG and emphasizes the flexibility of this assay to identify fusions without knowledge of the exact breakpoint-positions. To uncover why NanoFG did not identify the *SS18-SSX1* fusion gene, we manually investigated the candidate locus in the IGV Browser[18]. The sequencing of the HS-SYII cell line resulted in very little throughput, on-target coverage (11x) (Fig. 3a) and relatively low breakpoint-spanning reads (6). As a result, only one fusion-spanning read was produced, which is below the filtering cut-off for fusion-supporting reads set for NanoFG (requirement of minimal two fusion-supporting reads). When adjusting the settings of NanoFG to one supporting read, the *SS18-SSX1* fusion was called (Fig. 3a, e), however, lowering the threshold of fusion-supporting reads requires manual validation if the fusion status is unknown to exclude false-positives (Supplementary Fig. 3B). Despite the low-throughput for the HS-SYII cell line, the assay resulted in a 68x fusion-specific fold-enrichment (Fig. 3b). This shows the ability of FUDGE to identify fusion genes irrespective of fusion partner or breakpoint-location from low-coverage nanopore sequencing data.

**Detection of fusion genes from tumor material.** To validate that FUDGE identifies fusion genes from tumor material and without prior knowledge of the breakpoint-location, we applied the assay to six tumor samples of different origins with known fusion

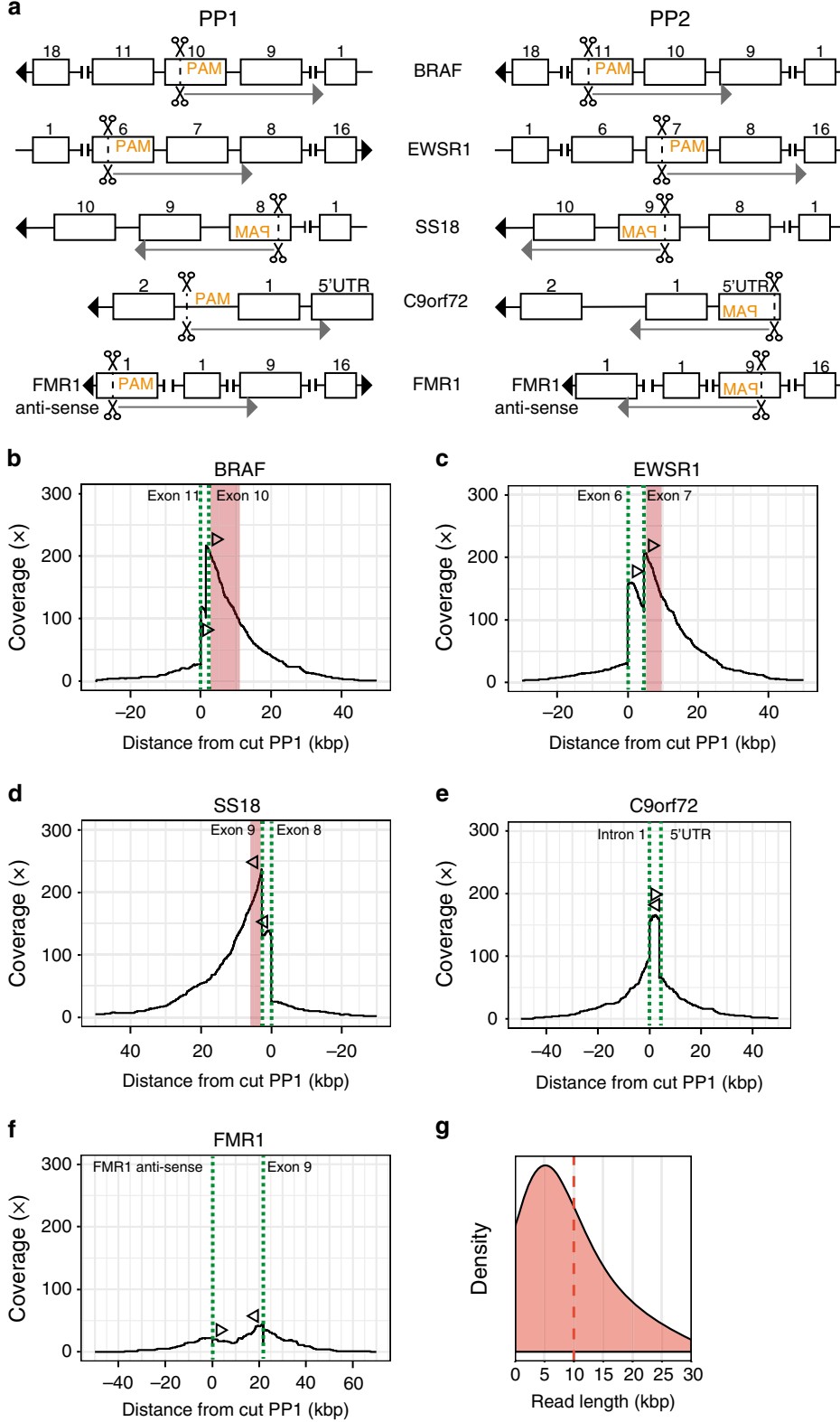

**Fig. 2 Cas9 enrichment across genomic loci. a** Cas9-introduced cuts targeting different genomic regions for crRNA pool PP1 and PP2. Boxes and numbers represent exons. Cut-positions are shown by scissors, PAM sequence (orange) indicates the directionality of crRNA design and arrows show the anticipated sequencing direction. **b–f** Coverage plots showing on-target coverage across multiple genomic loci. Dotted lines (green) indicate the crRNA-directed Cas9 cleavage positions and arrows indicate the directionality of reads created from the specific crRNA design. Red areas highlight the most common breakpoint-locations per gene. **g** The read-length distribution for the sequencing run. The dashed line indicates the mean read-length. Source data are provided as a Source Data file.

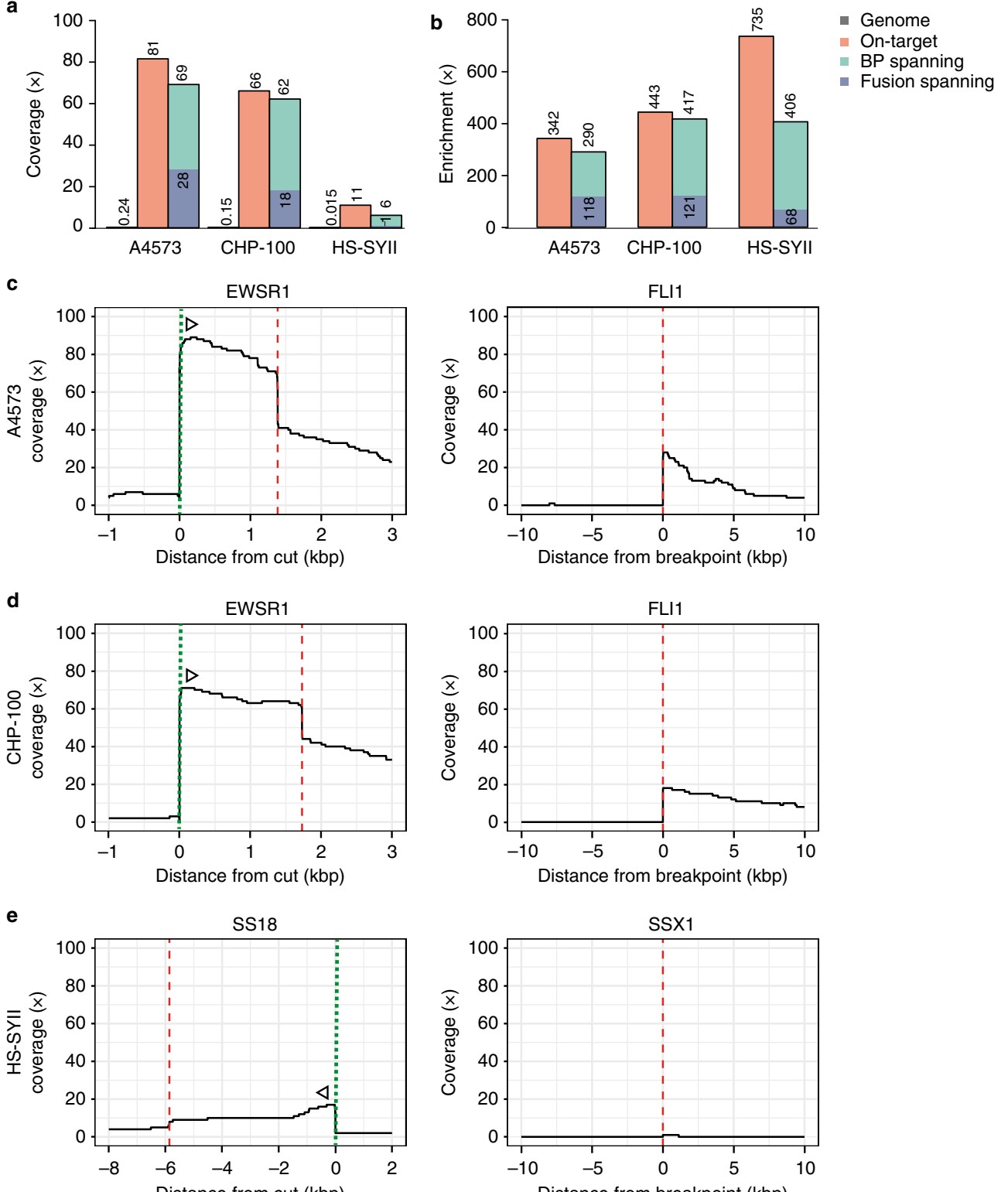

**Fig. 3 Coverage and enrichment across fusion-positive cancer cell lines. a** Mean coverage and **b** enrichment across the genome, on-target (cut to breakpoint), BP-spanning (breakpoint-spanning), and fusion-spanning (across the fusion junction) for the three cell lines A4573, CHP-100, and HS-SYII. **c** Coverage plots for the cell line A4573 for the two fusion partners *EWSR1* (targeted) and *FLI1*. **d** Coverage plots for the cell line CHP-100 for the two fusion partners *EWSR1* (targeted) and *FLI1*. **e** Coverage plots for the cell line HS-SYII for the two fusion partners *SS18* (targeted) and *SSX1*. Dotted lines (green) indicate the crRNA-directed Cas9 cleavage positions and dashed lines (red) indicate breakpoint positions. Arrows indicate the directionality of reads created from the specific crRNA design. Source data are provided as a Source Data file.

status. We tested DNA isolated from an Ewing sarcoma (ES1), a rhabdomyosarcoma (RH), a chronic myeloid leukemia (CML), a Burkitt's Lymphoma (BL), a philadelphia chromosome-positive B-lymphoblastic acute leukemia (B-ALL)(ALL1) and a B-ALL (ALL2). Rhabdomyosarcomas are characterized by breaks in the second intron of *FOXO1* (104 kb) which then fuses to either *PAX3* or *PAX7* (ref. [19]). Due to the large potential breakpoint region within *FOXO1*, we chose to target the *PAX3* and *PAX7* genes instead to minimize the number of necessary crRNAs. Here, the most common breakpoint areas span a 18 kb and 32 kb region, respectively. Therefore, we designed sequential crRNAs to span the potential breakpoint regions of both genes (Supplementary Data 1). The CML and the ALL1 harbored a *BCR-ABL1* fusion gene with unknown breakpoint position. The *BCR* gene harbors three recurrent breakpoint clusters, spanning 6.6 kb between exon 12 and exon 15 (major-cluster), 71 kb between exon 1 and exon 2 (minor-cluster), and 1.3 kb between exon 19 and exon 20 (micro-cluster). To comprehensively cover all possible breakpoints, we targeted all three clusters with in total eleven crRNAs (Supplementary Data 1). We sequenced each tumor sample on a single flow cell and identified, as expected, an *EWSR1-FLI1* fusion (ES1, 8 reads) (Supplementary Data 1 and Supplementary Figs. 3A and 4A), a *PAX3-FOXO1* fusion (RH, 32 reads) (Fig. 4a, d), a *BCR-ABL1* fusion within the major-cluster (CML, 22 reads) (Fig. 4b, d), a translocation between *MYC* and the *IGH* locus (BL, 3 reads) (Fig. 4d and Supplementary Fig. 4B), a *BCR-ABL1* fusion within the minor-cluster (ALL1, 27 reads) (Fig. 4d and Supplementary Fig. 4C) and a *CRLF2-P2RY8* rearrangement (ALL2, 185 reads) (Fig. 4c, d). The on-target enrichment was 498x (ES1), 930x (RH1), 611x (CML), 347x (BL), 679 (ALL1), and 3492 (ALL2) and the breakpoint-spanning enrichment was 406x (ES1), 838x (RH1), 598x (CML), 81x (BL), 633x (ALL1), and 3601x (ALL2) (Fig. 4e). From this, a fusion-specific enrichment of 270x (ES1), 258x (RH1), 188x (CML), 61x (BL), 197x (ALL1), and 3382x (ALL2) was achieved (Fig. 4e). Furthermore, we identified two additional fusion events, a reciprocal *FOXO1-PAX3* (RH2) fusion with eight fusion-supporting reads for the RH sample and a *DRICH1-BCR* (CML2) fusion with three fusion-supporting reads for the CML sample. As these events were unexpected findings, we validated them by breakpoint PCR (Supplementary Fig. 5A, B). We furthermore performed Sanger validation on the *DRICH1-BCR* fusion, as this event has not been previously reported in literature (Supplementary Fig. 5C). It is important to note that NanoFG is specifically designed to detect fusion genes with breakpoints within both of the involved fusion partners. As the *IGH/MYC* translocation (*IGH*-breakpoint approximatively 2.5 kb upstream of *IGHM*) and *CRLF2-P2RY8* rearrangement (*CRLF2*-breakpoint approximatively 3.5 kb upstream of *CRLF2*) do not meet this criterium, NanoFG does not report them and the use of NanoSV is more appropriate. For instances where a fusion event is expected in areas outside of annotated genes (including promoter, both UTRs, and exonic/intronic regions), manual analysis of the variant calling file (vcf) reported by NanoSV, an initial step in the NanoFG pipeline (Methods) is required. Here, the information on exact breakpoint position, number of supporting reads, etc. can be extracted.

In summary, this demonstrates the ability of FUDGE to detect known and reciprocal fusion genes and genomic rearrangements from patient samples irrespective of tumor type.

**Blinded fusion gene detection and run time analysis**. To confirm that FUDGE identifies fusion genes without prior knowledge of fusion partner or fusion status, we tested two tumor samples in a blinded manner (B1 and B2). For the B1 sample, diagnostic efforts identified a *KMT2A* fusion through break-apart FISH;

however, the fusion partner could not be identified and was unknown prior to the experiment described here. The *KMT2A* gene is a frequent fusion partner in AML and ALL and shows two major breakpoint clusters[4] of which we designed crRNAs for both (Supplementary Data 1). The B2 sample was randomly chosen out of a pool of six tumor samples (four ALL, one BL, one Burkitt's-ALL) which could potentially harbor a *BCR-ABL1*, *IGH/MYC*, or *CRLF2-P2RY8* rearrangement. Therefore, we targeted the B1 sample with two crRNAs and the B2 sample with 14 crRNAs (Supplementary Data 1) and sequenced both samples on one flow cell each. NanoFG identified a *KMT2A-MLLT6* fusion in B1 (Fig. 5a) and a *BCR-ABL1* fusion in B2 (Fig. 5b) with 29 fusion-spanning and 27 fusion-spanning reads, respectively (Fig. 5c). Overall, we observed a breakpoint-spanning enrichment of 938x (B1) and 313x (B2) and a fusion-spanning enrichment of 143x (B1) and 148x (B2) (Fig. 5d). This demonstrates the capacity of FUDGE to identify unknown fusion events from tumor material.

Furthermore, we performed a retrospective time-course experiment on all eight sequenced tumor samples to identify the necessary sequencing time to detect fusion-spanning reads (Fig. 5e, f). On-average, 70% of the fusion-spanning reads were produced within the first 12 h of sequencing and 90% of the fusion-spanning reads were produced within the first 24 h of sequencing (Fig. 5e). For all samples, except the IGH/MYC rearrangement in BL, it took less than three hours of sequencing time to identify two fusion-spanning read (Fig. 5f). This highlights the speed of our approach and indicates that if sequencing would be stopped after 24 h, the majority of fusion-spanning reads could be obtained.

**Fusion gene detection from low input tumor material**. The amount of available tumor material is often a limiting factor for genomic analysis. To circumvent this problem, we tested if our pipeline was compatible with whole genome amplified (WGA) material. WGA produces DNA fragments of considerable length (up to 100 kb)[20], and could therefore be a suitable method to produce enough DNA at sufficient length for targeted nanopore sequencing. Therefore, we sequenced WGA-DNA of two colon cancer samples (C1 and C2), known to harbor *BRAF* fusions (*AGAP3-BRAF* and *TRIM24-BRAF*, respectively)[21], a sarcoma sample with a *SS18-SSX1* fusion (S1) and a PDX sarcoma sample with unknown fusion status (S2). We targeted the S2 sample with nine crRNAs targeting the most common recurrent sarcoma fusion partners *EWSR1*, *PAX3*, *PAX7*, and *SS18*. For all samples we performed WGA on 10 ng starting material and subjected 1 µg of WGA-DNA to the enrichment protocol. Genome coverage (Fig. 6a) and read-length were comparable to previous experiments (Supplementary Data 1). Initially NanoFG did not detect the *AGAP3-BRAF* fusion, however, lowering the threshold to one fusion-supporting read identified the fusion gene (Fig. 6a–c). The *TRIM24-BRAF* fusion was called by NanoFG with eleven fusion-spanning reads (Fig. 6a, b, d). For the S1 and the S2 sample, neither NanoFG nor manual inspection in IGV could detect a targeted fusion gene. Notably, WGA introduced accompanying structural variation leading to a high number of fusion gene predictions (Supplementary Fig. 5D) and difficulties for manual inspection in IGV. However, we show that a fusion supporting threshold of two reads is a reasonable cut-off for normal and WGA-samples, as the number of predicted fusions decreases drastically compared to one supporting read but remains relatively stable compared to a higher fusion-support (Supplementary Fig. 5D). Furthermore, fusion genes identified by NanoFG that were not targeted through crRNAs within our assay are very likely to be false-positives. We successfully validated the two

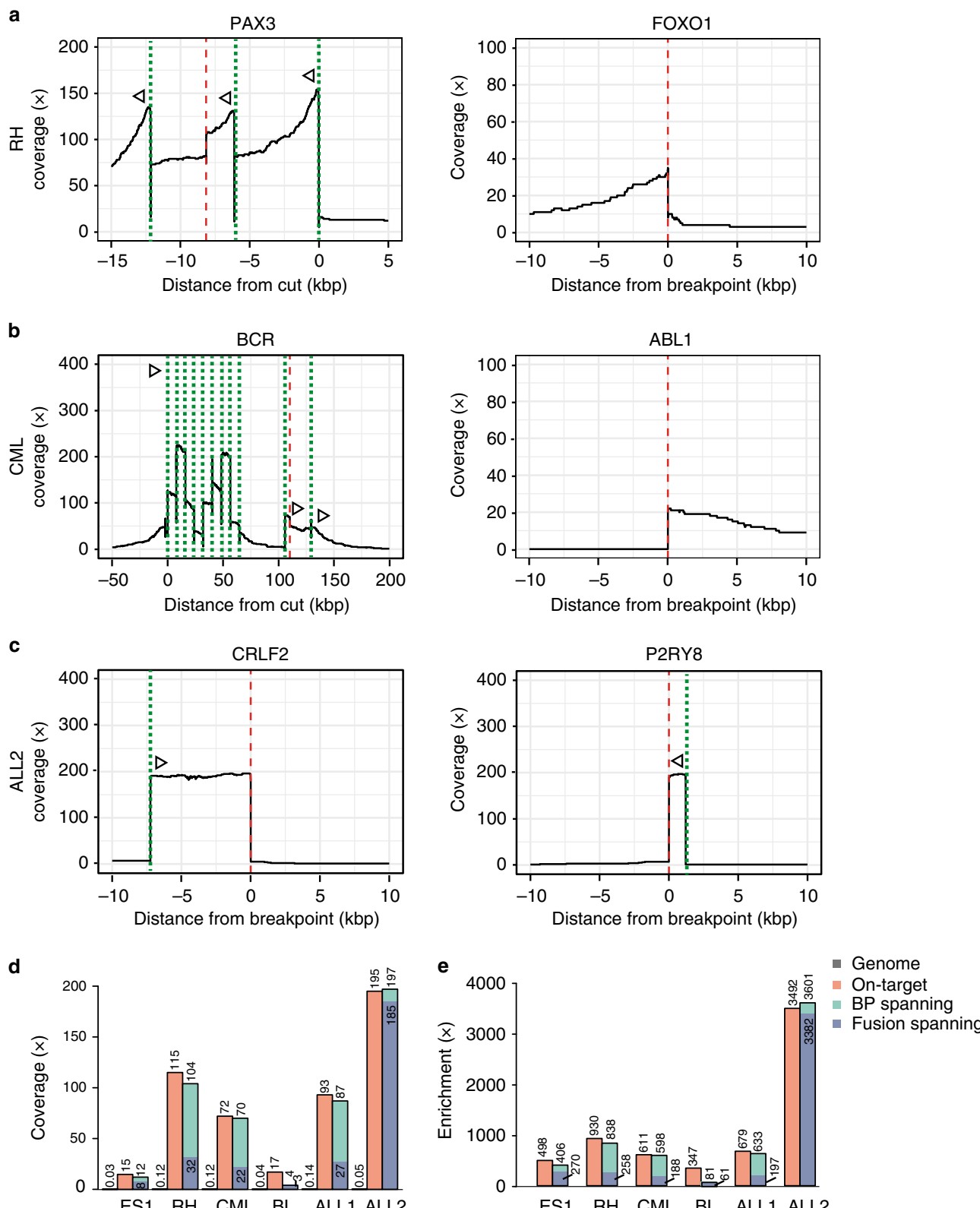

**Fig. 4 Fusion gene coverage and enrichment from tumor samples. a** Coverage plots for the RH tumor sample for the two fusion partners *PAX3* (targeted) and *FOXO1*. *PAX3* was targeted with three sequential guides to span the 18 kb possible breakpoint region. **b** Coverage plots for the CML tumor sample for the two fusion partners *BCR* (targeted) and *ABL1*. *BCR* was targeted with eleven sequential guides to span the possible breakpoint region. **c** Coverage plots for the ALL2 tumor sample for the deletion between *CRLF2* (targeted) and *P2RY8* (targeted). **d** Mean coverage and **e** mean enrichment across the genome, on-target (cut to breakpoint), BP-spanning (breakpoint-spanning), and fusion-spanning (across the fusion junction) for the tumor samples ES1, RH, CML, BL, ALL1, and ALL2. Dotted lines (green) indicate the crRNA-directed Cas9 cleavage positions and dashed lines (red) indicate breakpoint positions. Arrows indicate the directionality of reads created from the specific crRNA design. Source data are provided as a Source Data file.

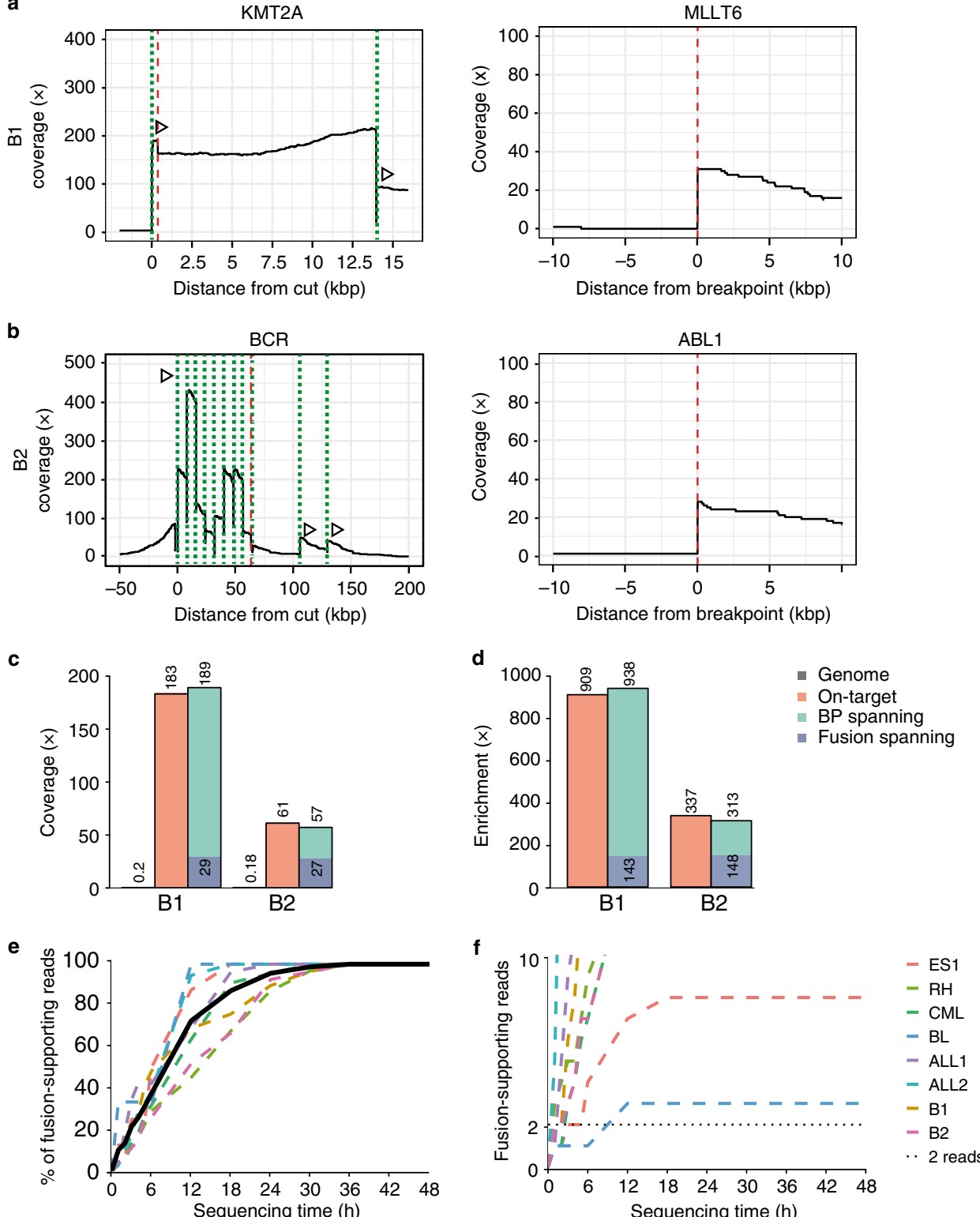

**Fig. 5 Fusion gene coverage and enrichment from blinded samples and run-time analysis. a** Coverage plots for the B1 tumor sample for the two fusion partners KMT2A (targeted) and *MLLT6*. **b** Coverage plots for the B2 tumor sample for the two fusion partners *BCR* (targeted) and *ABL1*. *BCR* was targeted with eleven sequential guides to span the possible breakpoint region. Dotted lines (green) indicate the crRNA-directed Cas9 cleavage positions and dashed lines (red) indicate breakpoint positions. Arrows indicate the directionality of reads created from the specific crRNA design. **c** Mean coverage and **d** mean enrichment across the genome, on-target (from cut to breakpoint), BP-spanning (breakpoint-spanning), and fusion-spanning (across the fusion junction) for the tumor samples B1 and B2. **e** Time-course experiment on sequencing time necessary to identify fusion-spanning reads (%) and **f** fusion-spanning reads (*n* = 2). Source data are provided as a Source Data file.

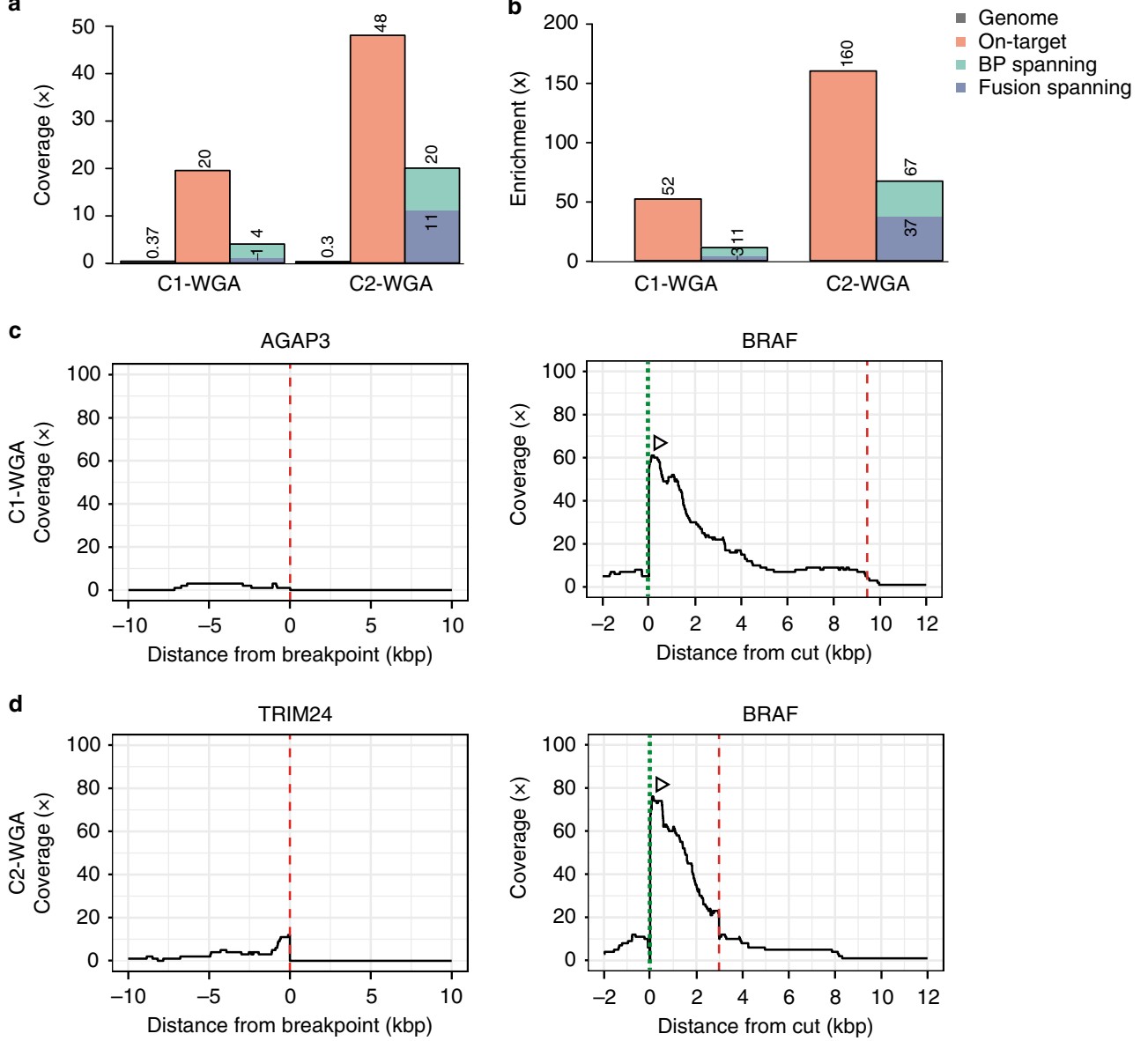

**Fig. 6 Fusion gene detection from WGA tumor DNA. a** Mean coverage and **b** enrichment across the genome, on-target (cut to breakpoint), BP-spanning (breakpoint-spanning), and fusion-spanning (across the fusion junction) for the whole genome amplified tumor material C1 and C2. **c** Coverage plots for the whole genome amplified tumor material C1 for the two fusion partners AGAP3 and BRAF (targeted). **d** Coverage plots for the whole genome amplified tumor material C2 for the two fusion partners TRIM24 and BRAF (targeted). Dotted lines (green) indicate the crRNA-directed Cas9 cleavage positions and dashed lines (red) indicate breakpoint positions. Arrows indicate the directionality of reads created from the specific crRNA design. Source data are provided as a Source Data file.

*BRAF* fusion genes, detected by a single fusion-spanning read (such as in A*GAP3-BRAF)*, by utilizing the exact breakpoint-locations provided by NanoFG and breakpoint-spanning PCR on the non-amplified tumor DNA (Supplementary Fig. 5E). In addition, for the *BRAF* fusions, the breakpoint junction locations were 6.5 kb apart (Fig. 6c, d and Supplementary Fig. 3), highlighting the unbiased performance of our assay. This demonstrates that FUDGE may be successfully applied to WGA material and NanoFG still accurately identifies the exact genomic breakpoint of the structural variants; however, prior knowledge of both fusion genes is required.

**Multiplexing of fusion positive cell lines**. Parallel identification and cost-reduction are key for diagnostic approaches. Therefore,

we tested the feasibility to multiplex samples in one sequencing run. We obtained DNA from four *KMT2A*-fusion positive cell lines (ALLPO, KOPN8, ML2 and Monomac-1) with different fusion partners (*MLLT1*, *MLLT2*, *MLLT3*, and *MLLT4*). We used two crRNAs targeting both breakpoint clusters (Supplementary Table. 1) and produced separate libraries for each sample (Fig. 7a). The targeted libraries were pooled pre-sequencing without barcoding and run on a single flow cell. This multiplexing approach resulted in a genome coverage of 0.57x and average read-length of 9.2 kb (Supplementary Data 1). NanoFG identified the four different fusion partners (Supplementary Fig. 6A) and 6 different breakpoint-locations (Fig. 7b). Interestingly, two *KMT2A*-fusions (*MLLT2* and *MLLT3*) appeared to be reciprocal (Supplementary Fig. 6A, B). The breakpoints within *KMT2A* spanned a region of 6 kb, and we identified breakpoints for

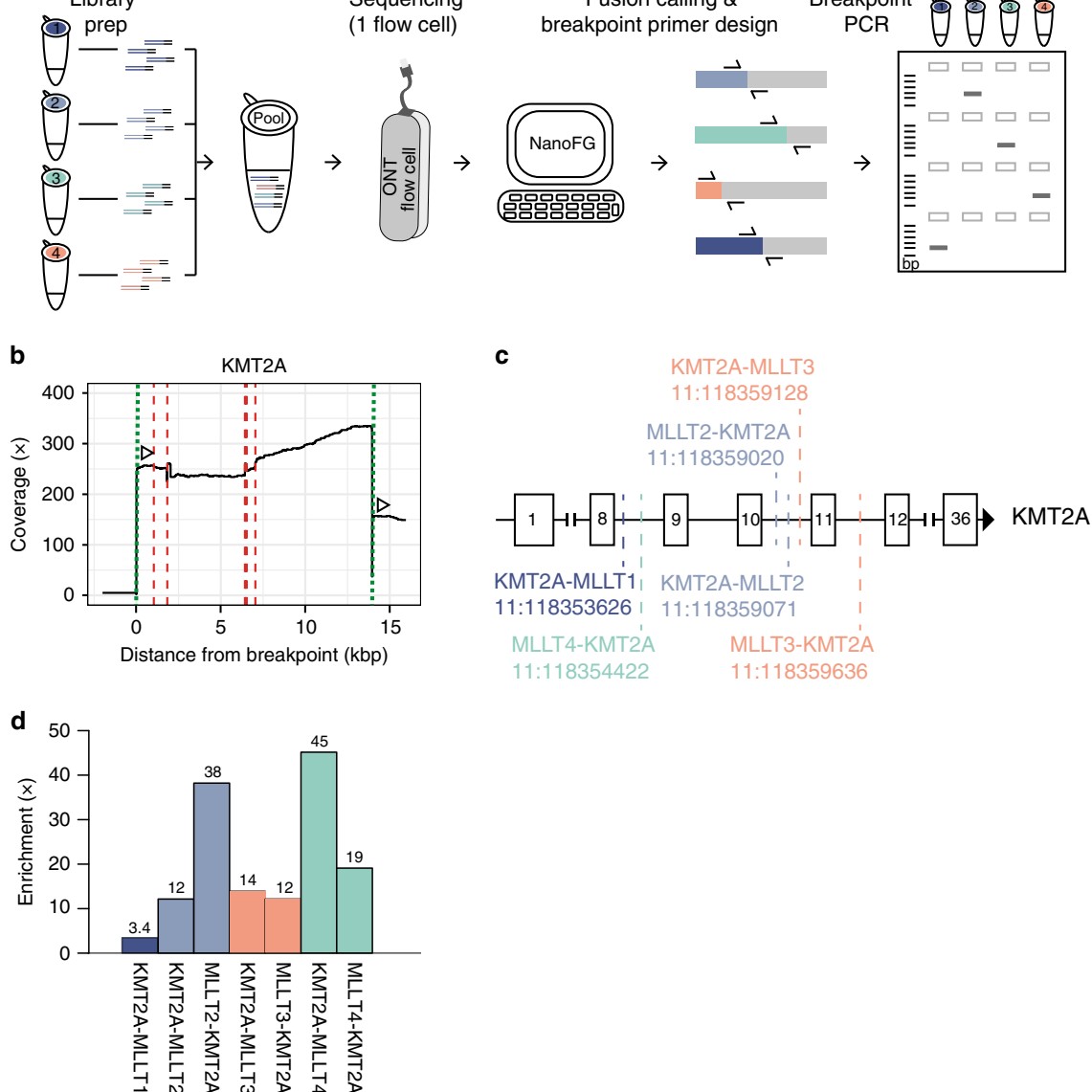

**Fig. 7 Multiplexing of fusion-positive samples with varying breakpoints and fusion partners. a** Schematic overview of the multiplexing approach. Samples are prepared and subjected to the Cas9-enrichment individually and pooled equally pre-sequencing. The library-pool is sequenced on a single ONT flow cell and NanoFG detected fusion genes and designed fusion-specific breakpoint primers. Original samples are subjected to breakpoint PCR to identify the sample-of-origin for each fusion gene. **b** Coverage plots for *KMT2A* (targeted). Dotted lines (green) indicate the crRNA-directed Cas9 cleavage positions and dashed lines (red) indicate breakpoint positions. Arrows indicate the directionality of reads created from the specific crRNA design. **c** Breakpoint locations within the *KMT2A* gene for the different identified fusion genes. Breaks cluster between Exon 8 and 12 and reciprocal fusion genes are highlighted in the same color. **d** Enrichment across the fusion junctions. Source data are provided as a Source Data file.

reciprocal fusions to be location-independent (Fig. 7c). We utilized the breakpoint-spanning primers and tested all samples for the occurrence of all fusion genes (Fig. 7a). This approach easily deconvoluted the sample-of-origin of each fusion, therefore validating this multiplexing approach (Supplementary Fig. 7A). Of note, the Monomac-1 cell line (*KMT2A-MLLT3*) also exhibited a positive result for the *KMT2A-MLLT1* fusion. This could be traced back to a contamination in the cultured cell line, highlighting the sensitivity of this assay to detect subclonal events. We isolated fresh DNA from the Monomac-1 cell line and could indeed only validate the expected fusion gene *KMT2A-MLLT3* (Supplementary Fig. 7B). Furthermore, from the coverage plot we observed 26 reads within the *MLLT4* fusion partner (Supplementary Fig. 6A) which were not explained by any of the NanoFG detected fusions. Upon manual investigation in the IGV browser,

we identified one fusion, *KMT2A-MLLT4*, that had a more complex rearrangement which was not called by NanoFG (Supplementary Fig. 7C). In this case, a small 30 bp region of *KMT2A* was deleted, followed by a 185 bp inversion and the ultimate fusion to *MLLT4*. We again designed breakpoint-spanning primers and in addition, performed Sanger-sequencing on the amplicons and validated the occurrence and structure of the complex rearrangement (Supplementary Fig. 7C). As a result, with the use of only one nanopore flow cell, we identified seven fusion genes from four samples with a collective on-target enrichment of 349x resulting in an average of 18 fusion-spanning reads (Fig. 7d). This shows the ability of our approach to multiplex samples with different fusion genes and breakpoint-positions and pinpoint the sample-of-origin by a simple PCR assay.

**Table 1 Comparison of DNA and RNA-based fusion gene detection methods.**

|  | DNA | | | | RNA | | |
|---|---|---|---|---|---|---|---|
| Stability | Stable | | | | Instable | | |
| Breakpoint | Across large genomic areas | | | | Only exon-exon junctions | | |
| Detection method | FISH | FUDGE | Targeted-NGS | WGS | RT-PCR | Targeted-NGS | RNA seq |
| Speed (days) | 1 | 2 | 7–14 | 7–14 | 1 | 7–14 | 7–14 |
| Targeted | Semi | Semi | Semi | No | Yes | Semi | No |
| Partner detection | No | Yes | Yes | Yes | N/A | Yes | Yes |
| Promoter fusions | No | Yes | Yes | Yes | No | No | No |
| Validation required | N/A | No | No | Yes (Sanger) | No | No | Yes (Sanger) |
| MRD | No | Yes | Yes | Yes | Yes | Suboptimal | Suboptimal |

## Discussion

Fusion genes are critical determinants for diagnosis, prognosis and treatment opportunities for various cancer types[22]. However, fusion gene detection by diagnostic approaches is limited to highly recurrent fusion gene configurations. We here developed FUDGE, a fusion detection assay from gene enrichment coupled to nanopore sequencing, which enables rapid partner-location and breakpoint-location independent fusion gene detection within 48 h.

Rapid identification of the genomic breakpoint offers the opportunity to utilize the breakpoint junctions as a biomarker for minimal residual disease (MRD) tracing[23]. Common diagnostic approaches for fusion gene detection can be divided into DNA or RNA-based approaches (Table 1). Detection of fusion genes on the RNA level might be less complex due to the restriction of breakpoints to exon-exon junctions; however, RNA molecules are less stable and the overall abundance is influenced by expression levels. DNA-based approaches such as targeted NGS assays or WGS are preferable since they identify all fusion gene events including promoter fusions, as well as the exact genomic breakpoint. However, these assays are hampered by longer turn-around times and WGS can result in high false-positive rates.

With FUDGE we offer fast and unbiased fusion gene detection. We successfully identified fusion genes from genomic DNA independent of cancer type or fusion gene configuration and/or breakpoint-positions. We targeted ten recurrent fusion partners within eight solid and hematological tumor specimens and identified 22 unique fusion gene configurations, highlighting the complexity of fusion gene biology. In one case, KMT2A was identified as a fusion partner by break-apart FISH through diagnostic efforts; however, the fusion partner was undetectable. We applied FUDGE to the sample and identified MLLT6 as the fusion partner within two days (provided the crRNA was already designed and in-house). Furthermore, FUDGE also detects reciprocal fusion events without additional efforts. In the case of two BRAF fusion-positive samples, the breakpoint locations were >6 kb apart from each other. Conventional methods such as qPCR would have not sufficed to span this large region of possible breakpoint-positions. We integrated an adaptation to the protocol to design sequential guides, offering the opportunity to span large regions of possible breakpoint-locations. For the BCR-ABL1 fusion, we spanned a >80 kb region, highlighting the versatility of FUDGE.

With our assay, fusion detection is possible within 48 h. Rapid identification of fusion genes is essential for tumor types where fusion genes are pathognomonic such as Ewing sarcoma or synovial sarcoma[22,24]. Hence, early detection allows for early definitive diagnosis and treatment initiation. Furthermore, occurrence of a specific fusion gene configuration can be a determinant of prognosis[25]. FUDGE identified all fusion gene configurations within 48 h, allowing immediate diagnosis and treatment initiation. In addition, we show that 70% of the fusion-supporting reads are produced in the first 12 h of sequencing and that three hours of sequencing are sufficient to identify two fusion-spanning reads, offering the opportunity to reduce the assay time for urgent cases to less than a day. Until now, we focused our assay on ten different recurrent fusion genes; however, expanding the assay to any gene of interest is possible. Furthermore, rapid detection of the exact genomic breakpoint-positions opens the door to immediately identify patient specific targets to trace fusion molecules within circulating tumor DNA (ctDNA) from liquid biopsies and asses treatment responses by monitoring of minimal residual disease.

A limitation of this approach is the requirement of non-fragmented DNA. Applying the FUDGE crRNA protocol to FFPE material (the current standard for pathology procedures), will most likely fail to comprehensively identify fusion genes due to short read lengths derived from degraded FFPE DNA. An adaptation of the design strategy to regularly interspace crRNAs at short intervals may overcome this issue; however, this approach will drastically increase the assay costs per fusion gene. Furthermore, intratumoral heterogeneity and tumor purity are likely to influence the lower detection limits of our assay, and the use of WGA in situations of low DNA availability may be useful to accurately identify the breakpoint-location but only with prior knowledge of both fusion gene partners. We set a cut-off of at least two fusion-spanning reads to reliably detect a fusion gene without further validation. In general, we observed a decrease in on-target coverage for low throughput sequencing runs and/or more distal breakpoint events (Supplementary Fig. 8), suggesting that higher coverage of breakpoints can be obtained by guides placed closer to breakpoints. Notably, none of the sequenced DNA samples used in these experiments was specifically isolated for long read sequencing. Thus, optimizing the isolation method and therefore the length of the DNA molecules and/or incorporating the tiling approach will have a positive effect on detecting these more distal events. Here, two fusions were only detected with one fusion-spanning read each, requiring the manual validation of the fusion gene by breakpoint PCR. However, by incorporating a multi-crRNA approach and increased efforts from ONT to improve sequencing throughput, the performance of FUDGE is expected to improve. In addition, the latter would allow for higher capacities to multiplex samples, reducing costs of the assay further.

Our current multiplexing approach, with sample pooling and retrospective demultiplexing by breakpoint PCR, reduces cost but prolongs assay duration and increases the complexity of sample processing. With lower throughput flow cells, such as the ONT Flongle, individual samples could be run separately, without pooling and demultiplex-PCR, thus simplifying the workflow and lowering assay costs dramatically.

In conclusion, FUDGE identifies fusion genes irrespective of fusion partner or breakpoint-location from low-coverage nanopore sequencing. FUDGE overcomes various limitations of current diagnostic assays by multiplexing targets in a rapid, accurate assay and can be applied to detect fusion genes within 48 h. The application of this assay in the clinic could allow for rapid gene fusion detection to allow appropriate therapy initiation and identification of specific genetic targets for blood-based minimal residual disease tracing.

## Methods

**Cell lines and culture.** Ewing sarcoma cell lines (A4573, CHP-100) and synovial sarcoma cell line (HS-SYII) were cultured in 5% $CO_2$ in a humidified atmosphere at 37 °C in Dulbecco's modified medium (DMEM) (Thermo Fisher) supplemented with 10% fetal bovine serum (FBS) and antibiotics (100 U/ml penicillin and 100 μg/ml streptomycin). The absence of *Mycoplasma sp.* contamination was determined with a Lonza MycoAlert system. Cell lines were obtained in collaboration from Anton Henssen, Charité Berlin.

ALL cell lines ALL-PO and KOPN8 and AML cell lines ML2 and Monomac-1 were maintained as suspension cultures in RPMI-1640 medium (Invitrogen), supplemented with 10% or 20% fetal calf serum (FCS) and antibiotics. Cell lines were obtained in collaboration from Ronald Stam, PMC Utrecht.

**Patient material.** The healthy donor (PP) provided written informed consent. The patients ES1 and RH had been registered and treated according to German trial protocols of the German Society of Pediatric Oncology and Hematology (GPOH). This study was conducted in accordance with the Declaration of Helsinki and Good Clinical Practice, and informed consent was obtained from all patients or their guardians. Collection and use of patient specimen were approved by the institutional review boards of Charité Universitätsmedizin Berlin. Specimen, clinical data were archived and made available by Charité-Universitätsmedizin Berlin.

C1 and C2 were previously sequenced[21] and were kindly provided by Prof Ijzermans, Dept of Surgery, Erasmus Medical Center Rotterdam, The Netherlands.

B1 was a kind gift from Prof. dr. C.M. Zwaan, Erasmus Medical Center—Sophia Children's Hospital, Rotterdam, The Netherlands/Princess Maxima Center for Pediatric Oncology, Utrecht, The Netherlands. Informed consent is given by the patient or his/her parents or legal guardians, and all is performed in line with the declaration of Helsinki, and the Erasmus MC—Sophia Children's Hospital approved the experiments.

CML, BL, ALL1, ALL2, and B2 were from the diagnostic sample archive of the Princess Máxima Center for Pediatric Oncology, Utrecht, The Netherlands. As the work was interpreted as falling within the scope of diagnostic service improvement, it did not require specific research ethics committee approval as stated in the EU Clinical Trials Directive (2001/20/EC).

**DNA Isolation.** Genomic DNA from cultured cells (A4573, CHP-100 and HS-SYII) and tissue (ES1 and RH) was extracted by using the column-based NucleoSpin® Tissue DNA extraction kit (Macherey-Nagel) following manufacturer's instructions. Sample quality control was performed using a 4200 TapeStation System (Agilent), and DNA content was measured with a Qubit 3.0 Fluorometer (Thermo Fisher).

Genomic DNA from the ALL cell lines (ALLPO and KOPN8), AML cell lines (ML2 and Monomac-1) and AML patient (B1) was isolated by using the column-based Qiagen DNeasy Blood and Tissue DNA extraction kit (Qiagen) following the manufacturer's instructions and DNA content was measured with a Qubit 2.0 Fluorometer (Thermo Fisher).

Genomic DNA was extracted either manually or with the QIAcube automated sample processor with the AllPrep DNA/RNA mini kit for CML, BL, and ALL2 and the QIAamp blood mini kit for the samples ALL1 and B2.

**WGA.** For whole genome amplification (WGA), 10 ng starting material was amplified with the repli-g mini kit (Qiagen) according to the manufacturer's protocol.

**crRNA design.** Each potential gene fusion constituted a known fusion partner to be targeted with this enrichment technique, and an (un)known partner to be identified following subsequent sequencing. The known target fusion partners were designated as a 5′ or 3′ fusion partner, dependent upon known literature. Furthermore, the most common breakpoint locations were extracted from a literature search and the most distal breakpoint locations were noted as extreme borders of the targeted area. If the unknown fusion partner was the 5′ partner, crRNAs were designed as the sequence present on the minus strand of the gene (5′–>3′) until the PAM sequence. If the unknown fusion partner was the 3′ partner, crRNAs were designed as the sequence present on the plus strand of the gene (5′–>3′) until the PAM sequence (Supplementary Fig. 1). Custom Alt-R® crRNAs were designed with

the Integrated DNA Technologies (IDT) custom gRNA design tool and chosen with maximum on-target and lowest off-target scores (IDT).

**Cas9 Enrichment and Nanopore Sequencing.** Cas9 enrichment was adapted from the ONT Cas9 enrichment protocol[12]. In brief, approximately 1 μg of genomic DNA or WGA-DNA (Supplementary Data 1) was dephosphorylated with Quick calf intestinal phosphatase (NEB) and CutSmart Buffer (NEB) for 10 min at 37 °C and inactivated for 2 min at 80 °C. crRNAs were resuspended in TE pH7.5 to 100 μM. For simultaneous targeting of multiple loci, crRNAs were pooled equimolarly to 100 μM. Ribonucleoprotein complexes (RNPs) were prepared by mixing 100 uM equimolarialy pooled crRNA pools with 100 μM tracrRNA (IDT) and duplex buffer (IDT), incubated for 5 min at 95 °C and thereafter cooled to room temperature. 10 μM RNPs were mixed with 62 μM HiFiCas9 (IDT) and 1× CutSMart buffer (NEB) and incubated at RT for 15 min to produce Cas9 RNPs. Dephosphorylated DNA sample and Cas9 RNPs were mixed with 10 mM dATP and Taq polymerase (NEB) at 37 °C for 15 min and 72 °C for 5 min to facilitate cutting of the genomic DNA and dA-tailing. Adapter ligation mix was prepared by mixing Ligation Buffer (SQK-LSK109, ONT), Next Quick T4 DNA Ligase (NEB) and Adapter Mix (SQK-LSK109, ONT). The mix was carefully applied to the processed DNA sample without vortexing and incubated at room temperature for 25 min. DNA was washed and bound to beads by adding TE pH8.0 and 0.3× volume AMPure XP beads (Agencourt) and incubated for 10 min at room temperature. Fragments below 3 kb were washed away by washing the bead-bound solution twice with Long Fragment Buffer (SQK-LSK109, ONT). Enriched library was released from the beads with Elution Buffer (SQK-LSK109, ONT). Enriched library concentration was measured with a Qubit Fluorometer 3.0 (Thermo Fisher).

The library from one tumor sample was loaded onto one flow cell (R 9.4, ONT) according to the manufacturer's protocol. Sequencing was performed on a GridION X5 instrument (ONT) and basecalling was performed by Guppy (ONT).

**NanoFG.** NanoFG can be found at https://github.com/SdeBlank/NanoFG.

Reads were mapped to the human reference genome version GRCHh37 by using minimap2 (v. 2.6)[26] with parameters: '-x map-ont -a'. The produced SAM file was compressed to bam format and indexed with samtools (v. 1.7)[27]. Next, structural variations were detected from the bam file. The user can choose either NanoSV (v. 1.2.4)[10] with default parameters: 'min_mapq=12, depth_support=False, mapq_flag=48', cluster_distance=100, ci_flag=300' or Sniffles (v.1.0.9)[28] with default parameters: '-s 2 -n -1 --genotype' to detect SVs. We here used NanoSV for all experiments (except multiplexing). For the samples C1 and HS-SYII, additional parameters: 'cluster_count=1' were used for NanoSV due to the low number of reads spanning the fusion. For the multiplexing experiment, the fraction of reads supporting the fusions was below the allele frequency cut-off in NanoSV. Therefore, the default Sniffles settings were used to detect 6 fusions. By default, all SVs that do not pass the built-in NanoSV or Sniffles filters are removed. In addition, all insertions are also removed from the VCF.

NanoFG selected candidate SVs that possibly form a fusion gene by annotating both ends of an SV with genes from the ENSEMBL database[29]. If both ends of the SV are positioned in different genes it was flagged as a possible fusion. Next, all the reads supporting the candidate SVs were extracted with samtools (v. 1.7)[27].

To remap and accurately detect SVs, all reads extracted per candidate fusion gene were re-mapped using LAST[30] (921) with default settings for increased mapping accuracy. Then, NanoSV was used to accurately define the breakpoints in the remapped fusion candidates. NanoSV parameters 'cluster_count=2, depth_support=False', cluster_distance=100, ci_flag=300' were used to detect all present fusions. For C1 and HS-SYII, 'cluster_count=1' was used as a parameter for NanoSV.

To check and flag fusions, additional information from the ENSEMBL database was gathered to produce an exact composition of the fusion gene. Only fusions that have the ability to produce a continuous transcript on the same strand were retained and additional flags were added to the sample to give extra indication if reported fusions are likely important or if some information from the ENSEMBL database is incomplete.

All gathered ENSEMBL gene information was used to produce an overview of the detected fusions. This includes the genes involved, the exon or intron containing the breakpoint, the exact position of the fusion, the number of fusion-supporting reads, involved CDS length of both fused genes and the final fused CDS length. The detected fusions were also reported in VCF format for further analysis. The number of fusion-supporting reads in the overview can differ from the number of reads reported in the vcf due to the fact that a read which supports a breakpoint multiple times in NanoSV is detected as a single supporting read by NanoFG. To give a better overview of detected fusions, NanoFG also produced a visual overview in PDF format. Apart from information on the genes, flags, position and fusion supporting reads it also included the locations of protein domains to provide quick insight into what domain are involved in the fusion.

NanoFG automatically designed primers for fusion gene validation using primer3 (ref. [31]) with default settings, aiming for a 200–400 bp product. Table 2 contains all primer sequences used for validation of breakpoints.

The run time of NanoFG on +−25000 nanopore reads is approximately 20 min using a single thread. Detailed instructions including a test-set can be found on GitHub (https://github.com/SdeBlank/NanoFG).

**Table 2 List of breakpoint-spanning primers.**

| | | | |
|---|---|---|---|
| SS18 fwd | CTGCTTGCCTCAACTAGAAAC | SSX1 rev | ATGAGCTTTAAATGGCTTGG |
| PAX3 fwd | CAAGTCAAGACTGTGATAGGC | FOXO1 rev | CAACCCTTCCATGACTCTTC |
| FOXO1 fwd | CCATATTCCACCTGAAGAGC | PAX3 rev | GCTTATAATCAAAGGCAGTGG |
| BCR fwd | AACGAATGTTGTGGGAAGTC | ABL1 rev | GCCCAAGATTATGTCTCCAG |
| DRICH1 fwd | TCACCCACATGGTCTGTAAG | BCR rev | ATCTCCTCAGGGAGAGTGAC |
| AGAP3 fwd | AGCGCCTTCAGCGACTACT | BRAF rev | GCAGACAAACCTGTGGTTGA |
| TRIM24 fwd | CTGTGAGGACAACGCAGAAG | BRAF rev | GCAGACAAACCTGTGGTTGA |
| KMT2A fwd 1 | CCAGGGATCTGTCTACCTTG | MLLT2 rev | CGTTTAAACCCTCCCTATTTC |
| MLLT2 fwd | AAGTGCTGGGATTCAAGGTG | KMT2A rev 1 | TGCTAAGTGACCTAAGAGTGG |
| MLLT4 fwd | TTGTATCCCTTGACCATTTG | KMT2A rev 2 | ATGTGAGCCACCACACTATC |
| KMT2A fwd 2 | CCAGGAATTCAAGGCTGTAG | MLLT1 rev | CTAGCATGTGGAGGAGACAG |
| KMT2A fwd 3 | ATGGCAGTGGCATTAAGAG | MLLT3 rev | CTTAGGTCACTTAGCATGTTCTG |
| MLLT3 fwd | ATTCATACCCACACCCAAAG | KMT2A rev 3 | GTCTCGAACTCCTGGCTTC |
| KMT2A inversion fwd | TGTTCTAGCCTAGGAATCTGC | KMT2A inversion rev | GCAGCACAGTGACACACAG |
| KMT2A translocation fwd | ATGTGAGCCACCACACTATC | MLLT4 translocation rev | ACAGACAGACGGCAAAGAG |
| KMT2A inversion fwd | TGTTCTAGCCTAGGAATCTGC | MLLT4 translocation rev | ACAGACAGACGGCAAAGAG |

**Minimal sequencing duration experiment**. To detect differences in fusion gene detection based upon sequencing duration, all fastqs were merged and all reads were sorted based on the time of sequencing. The earliest time was taken as the start of the sequencing run and subsequently reads were selected based on bins of 1, 2, 3, 4, 5, 6, 12, 18, 24, 30, 36, 42, and 48 h after the first read had been sequenced. NanoFG was then run separately on every fastq by using default settings for every sample. Using this approach, the time points where at least 2 supporting reads of a fusion have been sequenced can be determined to define the minimal sequencing duration necessary for each sample to produce two fusion-spanning reads.

**Minimal supporting read cut-off**. To select a minimum number of supporting reads used in the detection of fusion genes, we ran NanoFG on a number of samples (CHP-100, ES1, C1-WGA, and C2-WGA) with a minimum of one supporting read. Thereafter, the number of fusions reported with a minimum of 1, 2, 3, 4, and 5+ supporting reads were counted.

**Reporting summary**. Further information on research design is available in the Nature Research Reporting Summary linked to this article.

## Data availability

Low coverage WGS Binary Alignment Map (BAM) files from nanopore sequencing are available through controlled access at the European Genome-phenome Archive (EGA), hosted at the EBI and the CRG (https://ega-archive.org), with accession number EGAS00001003964. Requests for data access will be evaluated by the UMCU Department of Genetics Data Access Board (EGAC00001000432) and transferred on completion of a material transfer agreement and authorization by the medical ethical committee of the UMCU to ensure compliance with the Dutch medical research involving human subjects act. The source data underlying Figs. 2–7 are provided as a Source Data file. The ENSEMBL database for genome build GRCh37 can be found at https://grch37.ensembl.org/index.html. Any other relevant data are available from the authors upon reasonable request. Source data are provided with this paper.

## Code availability

NanoFG requirements, readme, and pipeline are at https://github.com/SdeBlank/NanoFG. Source data are provided with this paper.

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

## Acknowledgements

We thank all members of the Kloosterman and van Haaften groups for fruitful discussions and support. The authors thank KWF for supporting C.S. and W.P.K. grant UU 2012-5710. This work was supported by funds from the Utrecht University to implement a single-molecule sequencing facility. We thank the Utrecht Sequencing Facility for the Nanopore Sequencing. The colon cancer samples were kindly provided by Prof Ijzermans, Department of Surgery, Erasmus Medical Center Rotterdam, The Netherlands. Miriam Guillen Navarro, Susan Arentsen-Peters, Heathcliff Dorado-Garcia and Victor Bardinet have kindly helped with providing the clinical samples and information.

## Author contributions

C.S., W.P.K., and G.M. conceived the study. C.S. and G.M. designed experiments, and C.S., T.V., and I.R. performed the experiments. L.W., R.C.G., A.G.H., and R.S. provided samples and clinical information for the study. S.B., J.E.V.-I and M.J.R. performed bioinformatic analysis. C.S. and S.B. analyzed data and C.S., G.H., and G.M. interpreted the data. C.S. wrote the manuscript, which was edited by W.P.K., E.E.V., G.H., and G.M. and reviewed by all authors.

## Competing interests

The authors declare no competing interests.
