## [Peer Review File · Nature Communications]

Reviewers' Comments:

Reviewer #1:

Remarks to the Author:

Stangl et. al., "Partner-independent fusion gene detection by multiplexes CRISPR/Cas9 enrichment and long-read Nanopore sequencing".

In this study, the authors use a CRISPR/Cas9 excision and enrichment to enrich for the detection of translocation structural variants that generate common fusion-genes in cancer using long-read nanopore sequencing. Within the study they provide analytical validation of the protocol (Identification of gene fusions in cancer cell lines), proof-of-principle detection in tumor samples, and also demonstrate compatibility with low sample input amounts, and multiplexing, which are key considerations for clinical implementation. I thought the study was mature, well documented and the text and the figures were clear and understandable throughout.

Major Comments.

1. The authors demonstrate the analytical validity of the method, which is a commendable technical achievement. One thing I note is that the authors routinely describe enrichment according to the enrichment of coverage at the CRISPR site. Whilst this enrichment indicates the efficacy of the CRISPR/Cas9 crRNA site, it does not fully correlate with the enrichment at the translocation site. It is noted that coverage shows a marked decay curve downstream to the site, meaning that enrichment of reads at a translocation site may not be correlated with the cut-site enrichment (see Figure 3a). It is really this enrichment of reads at the translocation site (target-Locus) that is the salient metric, and should be primarily reported in the abstract etc. Nevertheless, these translocation site enrichment values do look very encouraging (average of 116 reads crossed fusion junctions relative to expected genome coverage of 0.5x?).

It is noted that this target-locus metric partly depends on how far downstream the translocation sites are to the CRISPR/Cas9 site. In cases of genes with long introns, this can be many distal and less sensitive, whilst, short introns and proximal translocation events will be more sensitive. It would be useful if the authors can discuss or report how far downstream the translocation sites are to their CRISPR/Cas9 site (it is difficult to tell from the red bars in the figures), and whether (and how much) they think they could achieve improved sensitivity by further optimising the location of the crRNA sites within introns (such as the tiling of crRNAs they do for the PAX3 gene)?

2. Within the study, the authors compare the advantages of the technique to alternative FISH and RT-PCR techniques. However, NGS fusion gene panels are now being routinely used in cancer diagnosis, and it would seem comparison to this approach is also fair. For example, fusion-gene panels can also detect novel gene partners, however, they cannot achieve the rapid turn-around times achieved by the authors. Could the authors include discussion of these alternative approaches within their study?

4. There are also substantial distinctions between the detection of fusion genes from RNA, rather than the translocation events in the DNA (as performed by the authors). I suspect that the detection of translocation events are more variable than detection of a fusion exon junction. However, conversely, the stability of DNA is beneficial in a clinical setting. It would be very helpful if the authors could perform long read nanopore sequencing of RNA extracted from the matched cell lines and thereby perform enable a more direct comparison of different considerations and diagnostic performance of the two approaches (RNA or DNA) that seem fundamental to the different performance parameters of this technique to qRT-PCR and fusion gene panels?

4. Whilst the authors do provide sound analytical validation of the approach, the study still requires validation for clinical application (although this is beyond the scope of this study). Whilst the use of DNA as an input and the rapid-turnaround times confer substantial advantages in a

clinical laboratory or hospital, and the authors perform some initial validations in patient samples, further work across a large patient cohorts are required to understand clinical reliability, utility, variability etc. Whilst beyond the scope of this study, I would encourage the authors to evaluate the approach across a wide range of different sample types, input amounts and ideally FFPE treated samples (which may fragment the DNA and thereby reduce the sensitivity of the method substantially)?

5. One additional concern is that the DNA input amount requirements seems quite high, particularly for clinical applications. Whilst the authors use whole genome amplifications, to they have a sense of the size of the amplified DNA fragments as this would presumably put an upper bound on the downstream distance that can be test?

Reviewer #2:

Remarks to the Author:

In this report, the authors present a method for enrichment of translocation breakpoint events followed by sequencing using the Oxford Nanopore system, as well as application of that method to several specimen types and fusion events. This marks the first report of such a system, to my knowledge, and enhances the suite of possible mechanisms to examine such events in patients with both benign and malignant neoplastic disease. The technology represents an advance from prior reports that employed nanopore based systems, which detail the identification of such events using whole genome sequencing, a necessarily more costly and likely long turnaround time approach (see e.g. Hang Au et. Al. in *Cancer Genetics*, 2019 <https://doi.org/10.1016/j.cancer.2019.08.005>).

However what this technique gains in targeting and enrichment, it loses in clearly stated clinical utility. The method, as described, targets a very small handful of targets. This makes it questionable whether it is diagnostically superior to FISH testing, which is inexpensive and widely used. Were the assay broader, that breadth could be the proposed benefit, but that is not claimed here. Moreover, the technique appears to be quite expensive. It uses a \$1000 flow cell to identify one fusion event if not multiplexed, and when used with multiple specimen testing, the assay requires the design and implementation of distinct PCRs for each fusion event identified. This translates to designing and waiting for primers, running the relevant PCRs and then a step to be verified by Sanger sequencing (which would almost certainly be needed in a clinical setting, where the stakes are too high to simply rely on getting a band product of the correct size). These considerations add to assay complexity, cost, and turnaround time. It is unclear whether the assay will work if sample barcoding is done prior to sequencing, due to the low number of reads being used to call some fusion events. If reads are misassigned to the wrong specimen, due to e.g. sequencing error, it will result in an unreliable assay. If proposed, such a method would need to be tested, preferably on blinded specimens. In which the researchers are blinded to the results. Moreover, the authors may be either unaware or are underplaying the widely used tests currently used to identify fusion events using knowledge of only a single partner. RNAseq and genome sequencing are mentioned, but there are a suite of other techniques that use enrichment to substantially lower costs while targeting a wide array of targets, and these often only require foreknowledge of one of the fusion partners. These include targeting RNA (see e.g. AMP sequencing, currently commercially offered by Archer Diagnostics), or DNA (e.g. Foundation One CDx, which uses biotinylated capture bait tiling of introns to detect fusions). These are the most meaningful competitors to the method described, and no direct comparison is made.

Despite that I think that this article, revised and with additional testing, merits publication for two reasons that are insufficiently highlighted by the authors. First, their method enables precise breakpoint detection and would offer an exquisitely sensitive test for minimal residual disease (MRD) in patients with fusion related tumors. The authors mention this application in a single sentence at the tail end of their discussion, but it is, in my opinion, the cornerstone of what would make this a useful technique without any changes to the assay (such as more targets or a more reasonable multiplexing technique). Second, their technique offers a molecular method for

identification of translocations that do not produce a fusion gene product and occur in difficult to sequence regions. IGH gene translocations are the first that come to mind. In these application the authors could actually reasonably be said to have provided an “unparalleled opportunity for pan-cancer detection of fusion genes”.

I would recommend this paper for publication with the following concerns addressed.

Major concerns:

1) None of the testing described appears to have been done in a blinded context. This, in addition to the relatively small number of specimens examined, limit the claims made in the paper regarding assay utility. Additional testing, ideally on primary patient specimens rather than cell lines, would enhance justification of the claims made in the abstract and discussion.

2) As stated above, the assay appears most important in its ability to detect the DNA breakpoints for translocations occurring in solid tumors. Few sites are tested, and one (SS18-SSX), appeared to work poorly. Additional successful testing in these tumor types would be most demonstrative of the assay's value.

3) The assay appears to require fresh specimens, though this does not appear to be explicitly stated. This is a MAJOR limitation in the diagnostic context, particularly as it applies to solid tumor excisions and biopsy specimens, and would be the deathknell for the technique if it merely identified the fusion genes, and not the precise genomic breakpoints, as most assays do. The authors need to highlight this in their discussion.

4) There is no justification given for the two read cutoff for calling fusion genes. Were fusion negative specimens tested to arrive at this number? Is this sufficiently specific? It's very difficult to evaluate specificity from the current presentation of the data, which currently focuses only on sensitivity. Two reads is an unusually low threshold for a sequencing related test, and may be unreasonable in the setting of WGA.

5) A more clinically applicable implementation of multiplexing, if performed, would be helpful. If not the discussion regarding the ease of deconvolution of pooled specimens is, I think, overstated. Ideally testing should be performed on patient specimens, not cell lines.

6) Finally, there are two areas (mentioned above) where this assay has immense promise, but that promise is not explored. These are in MRD detection and detection of non-fusion-producing translocations. Any material exploration of these would substantiate the authors' claims of clinical utility.

Minor concerns:

1) The format of the flow cell should be mentioned in the main body of the paper, as there is over an order of magnitude range of costs for a “nanopore flow cell” at this time. It should be clear that this describes work with the R9.4 and not flongle flow cell (which would be more impressive and more clinically plausible). This has major cost implications.

2) Some brief examination of the run time of NanoFG pipeline should be described. Apologies if I have missed it.

3) The authors need to make it clear in the introduction and discussion that this does not comprise the only method for targeted analysis of fusion breakpoints using high throughput sequencing. Many of these methods similarly have a fairly rapid turnaround time, show a higher degree of target enrichment, have a broader target panel, and have been demonstrated and are regularly used in a clinical setting. To the uninformed, it currently reads as if such options do not exist.

4) The paper mentions that false positives were identified when performing WGA prior to NanoFG. Additional data regarding the rate of these false positives should be given and it should be stated whether or not any false positives occurred in the target area (it appears that they were not, but it's not explicitly stated).

We thank both reviewers for their critical review and time. Please find our responses below.

Reviewer #1 (Remarks to the Author):

Stangl et. al., “Partner-independent fusion gene detection by multiplexed CRISPR/Cas9 enrichment and long-read Nanopore sequencing”.

In this study, the authors use a CRISPR/Cas9 excision and enrichment to enrich for the detection of translocation structural variants that generate common fusion-genes in cancer using long-read nanopore sequencing. Within the study they provide analytical validation of the protocol (Identification of gene fusions in cancer cell lines), proof-of-principle detection in tumor samples, and also demonstrate compatibility with low sample input amounts, and multiplexing, which are key considerations for clinical implementation. I thought the study was mature, well documented and the text and the figures were clear and understandable throughout.

Major Comments.

1. The authors demonstrate the analytical validity of the method, which is a commendable technical achievement. One thing I note is that the authors routinely describe enrichment according to the enrichment of coverage at the CRISPR site. Whilst this enrichment indicates the efficacy of the CRISPR/Cas9 crRNA site, it does not fully correlate with the enrichment at the translocation site. It is noted that coverage shows a marked decay curve downstream to the site, meaning that enrichment of reads at a translocation site may not be correlated with the cut-site enrichment (see Figure 3a). It is really this enrichment of reads at the translocation site (target-Locus) that is the salient metric, and should be primarily reported in the abstract etc. Nevertheless, these translocation site enrichment values do look very encouraging (average of 116 reads crossed fusion junctions relative to expected genome coverage of 0.5x?).

Answer: We thank the reviewer for the comment and agree that reporting the enrichment at the translocation-site as opposed to the enrichment at the cut-site should be primarily reported in the abstract. We have changed the abstract and main text accordingly and furthermore introduced a clearer distinction between breakpoint-spanning (reads that span from the cut across the breakpoint, including fusion-spanning reads) and fusion-spanning reads (reads that span from cut and continue within the fusion partner) to offer more clarity.

Textual changes

Abstract, page 2:

“We observe on average a 665 fold enrichment at the breakpoint-site and identify fusion breakpoints at nucleotide resolution - all within two days.”

Introduction, page 4:

“Utilizing this approach, we are able to achieve an average breakpoint-spanning coverage of 68x - resulting in an average enrichment of 665x - and identify fusion gene partners from various cancer types (e.g. AML, Ewing Sarcoma, Colon) within 48 hours.”

2. It is noted that this target-locus metric partly depends on how far downstream the translocation sites is to the CRISP/Cas9 site. In cases of genes with long introns, this can be many distal and less sensitive, whilst, short introns and proximal translocation events will be more sensitive. It would be useful is the authors can discuss or report how far downstream the translocation sites are to their CRISPR/Cas9 site (it is difficult to tell from the red bars in the figures), and whether (and how much) they think they could achieve improved sensitivity by further optimising the location of the crRNA sites within introns (such as the tiling of crRNAs they do for the PAX3 gene)?

Answer: We agree that the distance between the cut-site and breakpoint influences the sensitivity of the approach. On top of that, the read length and tumor purity are contributors to the success of the protocol. It has to be noted that none of the sequenced DNA samples were specifically isolated for long-read sequencing analysis or selected for high tumor purities but still we identified the fusion genes in all cases. However, by selecting for high tumor purities and by applying DNA isolation methods that protect the integrity of the DNA, the sensitivity of the assay could presumably be increased. To show the relation between genomic distance and sensitivity (coverage), we have added information on the distance from cut-site to breakpoint in the Supplementary Table 1 as well as a Figure (Supplementary Figure 8) depicting the correlation, and emphasized in the text that optimal crRNA design as close as possible (or with multiple guides with shorter distances between them) to the breakpoint is necessary for optimal performance of FUDGE.

Textual changes

Discussion, page 21:

*“In general, we observed a decrease in on-target coverage for low throughput sequencing runs and/or more distal breakpoint events (**Supplementary Figure 8**), suggesting that higher coverage of breakpoints can be obtained by guides placed closer to breakpoints. Notably, none of the sequenced DNA samples used in these experiments was specifically isolated for long-read sequencing. Thus, optimizing the isolation method and therefore the length of the DNA molecules and/or incorporating the tiling approach will have a positive effect on detecting these more distal events”*

Supplemental Figure 8:

Plot depicting the relationship between coverage at the target-locus, distance between the cut and the breakpoint, and sequencing throughput in Gbs.

3. Within the study, the authors compare the advantages of the technique to alternative FISH and RT-PCR techniques. However, NGS fusion gene panels are now being routinely used in cancer diagnosis, and it would seem comparison to this approach is also fair. For example, fusion-gene panels can also detect novel gene partners, however, they cannot achieve the rapid turn-around times achieved by the authors. Could the authors include discussion of these alternative approaches within their study?

Answer: We thank the reviewer for the comment. We have adapted the text and highlight now other targeted NGS approaches for fusion gene detection. Furthermore, we discuss the advantages and disadvantages of the different techniques with respect to our method (see also point 4 reviewer 1).

Textual changes

Introduction, page 3:

“Recently, next generation sequencing (NGS) assays which specifically target recurrent fusion partners have been developed and are currently implemented in clinical practice^{1,2}. These assays are highly versatile with respect to partner identification and input material (e.g. suitable for DNA isolated from Formalin-Fixed Paraffin Embedded tissue blocks; FFPE), but are accompanied with longer turnaround-times, increased costs and bioinformatic challenges.”

Discussion, page 20:

“Common diagnostic approaches for fusion gene detection can be divided into DNA or RNA-based approaches (Table 1). Detection of fusion genes on the RNA level might be less complex due to the restriction of breakpoints to exon-exon junctions; however, RNA molecules are less stable and the overall abundance is influenced by expression levels. DNA-based approaches such as targeted NGS assays or WGS are preferable since they identify all fusion gene events including promoter fusions as well as the exact genomic breakpoint. However, these assays are hampered by longer turn-around times and WGS can result in high false-positive rates.”

4. There are also substantial distinctions between the detection of fusion genes from RNA, rather than the translocation events in the DNA(as performed by the authors). I suspect that the detection of translocation events are more variable than detection of a fusion exon junction. However, conversely, the stability of DNA is beneficial in a clinical setting. It would be very helpful if the authors could perform long read nanopore sequencing of RNA extracted from the matched cell lines and thereby perform enable a more direct comparisons of different considerations and diagnostic performance of the two approaches (RNA or DNA) that seem fundamental to the different performance parameters of this technique to qRT-PCr and fusion gene panels?

*Answer: As pointed out by the reviewer, there are advantages and disadvantages between the detection of fusion genes on a RNA vs. DNA level. The genomic breakpoint detection is more complex but offers, as shown by us and others, opportunities to use the genomic breakpoint as a biomarker for reliable minimal residual disease (MRD) tracing from e.g. liquid biopsies ^{3,4}. The transcriptomic breakpoint detection is less complex (only exon-exon junctions are possible) but is restricted to expressed fusion transcripts and is suboptimal for MRD tracing due to the influence of expression levels on abundant molecules. Hence, we are opting for the genomic breakpoint to provide a genomic biomarker for MRD tracing. Performing long-read nanopore sequencing on RNA from a matched sample will, in our opinion, not add to the value or validation of the assay. Furthermore, there is no suitable enrichment protocol for recurrent fusion partners on RNA level that is compatible with nanopore sequencing and these technical differences would make a direct comparison impossible. We now highlight in the text our incentive to detect the genomic and not the transcriptomic breakpoint (**Table 1**).*

Textual changes

Discussion, page 20:

*“Common diagnostic approaches for fusion gene detection can be divided into DNA or RNA-based approaches (**Table 1**). Detection of fusion genes on the RNA level might be less complex due to the restriction of breakpoints to exon-exon junctions; however, RNA molecules are less stable and the overall abundance is influenced by expression levels. DNA-based approaches such as targeted NGS assays or WGS are preferable since they identify all fusion gene events including promoter fusions as well as the exact genomic breakpoint. However, these assays are hampered by longer turn-around times and WGS can result in high false-positive rates.”*

Table 1

	DNA	RNA
Stability	Stable	Unstable
Breakpoint	Across large genomic areas	Only exon-exon junctions

Detection method	FISH	FUDGE	Targeted-NGS	WGS	RT-PCR	Targeted-NGS	RNA seq
Speed (days)	1	2	7-14	7-14	1	7-14	7-14
Targeted	Semi	Semi	Semi	No	Yes	Semi	No
Partner detection	No	Yes	Yes	Yes	N/A	Yes	Yes
Promoter fusions	No	Yes	Yes	Yes	No	No	No
Validation required	N/A	No	No	Yes (Sanger)	No	No	Yes (Sanger)
MRD	No	Yes	Yes	Yes	Yes	Suboptimal	Suboptimal

4. Whilst the authors do provide sound analytical validation of the approach, the study still requires validation for clinical application (although this is beyond the scope of this study). Whilst the use of DNA as an input and the rapid-turnaround times confer substantial advantages in a clinical laboratory or hospital, and the authors perform some initial validations in patient samples, further work across a large patient cohorts are required to understand clinical reliability, utility, variability etc. Whilst beyond the scope of this study, I would encourage the authors to evaluate the approach across a wide range of different sample types, input amounts and ideally FFPE treated samples (which may fragment the DNA and thereby reduce the sensitivity of the method substantially)?

Answer: We thank the reviewer for the comment. In line with point 1 of reviewer 2 (major comments), we have performed further validations of the performance of our assay on clinical samples in a blinded context. Furthermore, we have expanded targets with respect to tumor type and recurrent gene partners (see point 1, reviewer 2). As for input DNA, FFPE material is usually fragmented to < 500 bps and thus not ideal for long read sequencing. Applying FUDGE to FFPE material could certainly be a possibility when designing a large panel of crRNAs per recurrent gene partner (e.g. a crRNA every 300 bps) to ensure that all genomic areas are covered. We think, however, that the cost-benefit ratio in this approach will not be high enough and that other methods more suitable for FFPE material should be the detection method of choice in these cases. In the manuscript we now highlight that our method optimally requires fresh frozen materials and other methods are available for more fragmented and degraded DNA specimen.

Textual changes

Results, page 5:

“To achieve this, genomic DNA isolated from fresh frozen samples is dephosphorylated as previously described⁵ and a crRNA flanking the suspected breakpoint region(s) is utilized to target Cas9 to a specific genomic loci where it creates a double-strand DNA break (Figure 1A).”

Discussion, page 21:

“A limitation of this approach is the requirement of non-fragmented DNA. Applying the FUDGE crRNA protocol to FFPE material (the current standard for pathology procedures), will most likely fail to comprehensively identify fusion genes due to short read lengths derived from degraded FFPE DNA. An adaptation of the design strategy to regularly interspace crRNAs at short intervals may overcome this issue; however, this approach will drastically increase the assay costs per fusion gene.”

5. One additional concern is that the DNA input amount requirements seems quite high, particularly for clinical applications. Whilst the authors use whole genome amplifications, to they have a sense of the size of the amplified DNA fragments as this would presumably put an upper bound on the downstream distance that can be test?

Answer: In this study we use multiple displacement amplification based whole genome amplification (WGA) that can be applied to very little DNA input (1-10 ng). This technique produces very large fragments of DNA (up to 100 kb)⁶ that are suitable for long-read sequencing approaches.

Textual changes

Results, page 16:

“WGA produces DNA fragments of considerable length (up to 100 kb)⁶, and could therefore be a suitable method to produce enough DNA at sufficient length for targeted nanopore sequencing.”

Reviewer #2 (Remarks to the Author):

In this report, the authors present a method for enrichment of translocation breakpoint events followed by sequencing using the Oxford Nanopore system, as well as application of that method to several specimen types and fusion events. This marks the first report of such a system, to my knowledge, and enhances the suite of possible mechanisms to examine such events in patients with both benign and malignant neoplastic disease. The technology represents an advance from prior reports that employed nanopore based systems, which detail the identification of such events using whole genome sequencing, a necessarily more costly and likely long turnaround time approach (see e.g. Hang Au et. Al. in Cancer Genetics, 2019 <https://doi.org/10.1016/j.cancergen.2019.08.005>).

However what this technique gains in targeting and enrichment, it loses in clearly stated clinical utility. The method, as described, targets a very small handful of targets. This makes it questionable whether it is diagnostically superior to FISH testing, which is inexpensive and widely used. Were the assay broader, that breadth could be the proposed benefit, but that is not claimed here. Moreover, the technique appears to be quite expensive. It uses a \$1000 flow cell to identify one fusion event if not multiplexed, and when used with multiple specimen testing, the assay requires the design and implementation of distinct PCRs for each fusion event identified. This translates to designing and waiting for primers, running the relevant PCRs and then a step to be verified by Sanger sequencing (which would almost certainly be needed in a clinical setting, where the stakes are too high to simply rely on getting a band product of the correct size). These considerations add to assay complexity, cost, and turnaround time. It is unclear whether the assay will work if sample barcoding is done prior to sequencing, due to the low number of reads being used to call some fusion events. If reads are misassigned to the wrong specimen, due to e.g. sequencing error, it will result in an unreliable assay. If proposed, such a method would need to be tested, preferably on blinded specimens. In which the researchers are blinded to the results.

Moreover, the authors may be either unaware or are underplaying the widely used tests currently used to identify fusion events using knowledge of only a single partner. RNAseq and genome sequencing are mentioned, but there are a suite of other techniques that use enrichment to substantially lower costs while targeting a wide array of targets, and these often only require foreknowledge of one of the fusion partners. These include targeting RNA (see e.g. AMP sequencing, currently commercially offered by Archer Diagnostics), or DNA (e.g. Foundation One CDx, which uses biotinylated capture bait tiling of introns to detect fusions). These are the most meaningful competitors to the method described, and no direct comparison is made.

Despite that I think that this article, revised and with additional testing, merits publication for two reasons that are insufficiently highlighted by the authors. First, their method enables precise breakpoint detection and would offer an exquisitely sensitive test for minimal residual disease (MRD) in patients with fusion related tumors. The authors mention this application in a single sentence at the tail end of their discussion, but it is, in my opinion, the cornerstone of what would make this a useful technique without any changes to the assay (such as more targets or a more reasonable multiplexing technique). Second, their technique offers a molecular

method for identification of translocations that do not produce a fusion gene product and occur in difficult to sequence regions. IGH gene translocations are the first that come to mind. In these application the authors could actually reasonably be said to have provided an “unparalleled opportunity for pan-cancer detection of fusion genes”.

I would recommend this paper for publication with the following concerns addressed.

Major concerns:

1) None of the testing described appears to have been done in a blinded context. This, in addition to the relatively small number of specimens examined, limit the claims made in the paper regarding assay utility. Additional testing, ideally on primary patient specimens rather than cell lines, would enhance justification of the claims made in the abstract and discussion.

Answer: We agree with the reviewer that further validation of different fusion genes and/or tumor types, preferably in a blinded context, would substantially strengthen the manuscript. We have now included an additional seven primary tumors and one PDX sample, targeting four new recurrent fusion gene partners (CRLF2, P2Y8, MYC and BCR) and three new tumor types (ALL, Burkitt Lymphoma and CML). The targeted fusion genes are commonly rearranged across various cancer types, thereby highlighting to the applicability of FUDGE. Furthermore, one of the additional experiments was performed in a blinded manner. Here, we prepared a pool of target crRNAs of possible fusions and chose a sample for sequencing unaware of the fusion status. In total we now applied FUDGE to seven cell lines, eleven tumors, and one PDX samples that were from eight different tumor types (hematological and solid). We targeted ten different fusion gene partners and identified 22 unique fusion gene configurations (or breakpoints).

Textual changes

Results, page 11-14:

Detection of fusion genes from tumor material

*“To validate that FUDGE identifies fusion genes from tumor material and without prior knowledge of the breakpoint-location, we applied the assay to six tumor samples of different origins with known fusion status. We tested DNA isolated from an Ewing sarcoma (ES1), a rhabdomyosarcoma (RH), a CML (CML), a Burkitt Lymphoma (BL), an ALL (ALL1) and an ALL patient with Down Syndrome (ALL2). Rhabdomyosarcomas are characterized by breaks in the second intron of FOXO1 (104 kb) which then fuses to either PAX3 or PAX7⁷. Due to the large potential breakpoint region within FOXO1, we chose to target the PAX3 and PAX7 genes instead to minimize the number of necessary crRNAs. Here, the most common breakpoint areas span an 18 kb and 32 kb region, respectively. Therefore, we designed sequential crRNAs to span the potential breakpoint regions of both genes (**Supplementary Table 1**). The CML and the ALL sample (ALL1) harbored a BCR-ABL1 fusion gene with unknown breakpoint position. The BCR gene harbors three recurrent breakpoint clusters, spanning 6.6 kb between exon 12 and exon 15 (major-cluster), 71 kb between exon 1 and exon 2 (minor-cluster), and 1.3 kb between exon 19 and exon 20 (micro-cluster). To comprehensively cover all possible breakpoints, we targeted all three clusters with in total eleven crRNAs (**Supplementary Table 1**). We sequenced each tumor sample on a single*

flow cell and identified, as expected, a EWSR1-FLI1 fusion (ES1, 8 reads) (**Supplementary Table 1 and Supplementary Figure 3 and 4A**), a PAX3-FOXO1 fusion (RH, 32 reads) (**Figure 4A and 4D**), a BCR-ABL1 fusion within the major-cluster (CML, 22 reads) (**Figure 4B and 4D**), a translocation between MYC and the IGH locus (BL, 3 reads) (**Figure 4D and Supplementary Fig. 4B**), a BCR-ABL1 fusion within the minor-cluster (ALL1, 27 reads) (**Figure 4D and Supplementary Figure C**) and a CRLF2-P2RY8 rearrangement (ALL2, 185 reads) (**Figure 4C and 4D**). The on-target enrichment was 498x (ES1), 930x (RH1), 611x (CML), 347x (BL), 679 (ALL1) and 3492 (ALL2) and the breakpoint-spanning enrichment was 406x (ES1), 838x (RH1), 598x (CML), 81x (BL), 633x (ALL1) and 3601x (ALL2) (**Figure 4E**). From this, a fusion-specific enrichment of 270x (ES1), 258x (RH1), 188x (CML), 61x (BL), 197x (ALL1) and 3382x (ALL2) was achieved (**Figure 4E**). Furthermore, we identified two additional fusion events, a reciprocal FOXO1-PAX3 (RH2) fusion with eight fusion-supporting reads for the RH sample and a DRICH1-BCR (CML2) fusion with three fusion-supporting reads for the CML sample. As these events were unexpected findings, we validated them by breakpoint PCR (**Supplementary Figure 5A and 5B**). We furthermore performed Sanger validation on the DRICH1-BCR fusion, as this event has previously not been reported in literature (**Supplementary Figure 5C**). It is important to note that NanoFG is specifically designed to detect fusion genes with breakpoints within both of the involved fusion partners. As the IGH/MYC translocation (IGH-breakpoint approximately 2.5kb upstream of the IGHM gene) and CRLF2-P2RY8 rearrangement (CRLF2-breakpoint approximately 3.5kb upstream of the CRLF2 gene do not follow this criteria, NanoFG does not report them and the use of NanoSV is more appropriate. For instances where a fusion event is expected in areas outside of annotated genes (including promoter, both UTRs, and exonic/intronic regions), manual analysis of the variant calling file (vcf) reported by NanoSV, an initial step in the NanoFG pipeline (**Methods**) is required. Here, the information on exact breakpoint position, number of supporting reads, etc can be extracted.

In summary, this demonstrates the ability of FUDGE to detect known and reciprocal fusion genes and genomic rearrangements from patient samples irrespective of tumor type.

Figure 4: Fusion gene coverage and enrichment from tumor samples

(A) Coverage plots for the RH tumor sample for the two fusion partners PAX3 (targeted) and FOXO1. PAX3 was targeted with three sequential guides to span the 18kb possible breakpoint region. (B) Coverage plots for the CML tumor sample for the two fusion partners BCR (targeted) and ABL1. BCR was targeted with eleven sequential guides to span the possible breakpoint region. (C) Coverage plots for the ALL2 tumor sample for the deletion between CRLF2 (targeted) and P2RY8 (targeted). (D) Mean coverage and (E) mean

enrichment across the genome, on-target (cut to breakpoint), BP-spanning (breakpoint-spanning), and fusion-spanning (across the fusion junction) for the tumor samples ES1, RH, CML, BL, ALL1 and ALL2. Dotted lines (green) indicate the crRNA-directed Cas9 cleavage positions and dashed lines (red) indicate breakpoint positions. Arrows indicate the directionality of reads created from the specific crRNA design. Source data are provided as a Source Data file.”

Results, page 14-16:

Blinded fusion gene detection and run-time analysis

“To confirm that FUDGE identifies fusion genes without prior knowledge of fusion partner or fusion status, we tested two tumor samples in a blinded manner (B1 and B2). For the B1 sample, diagnostic efforts identified a KMT2A fusion through break-apart FISH; however, the fusion partner could not be identified and was unknown prior to the experiment described here. The KMT2A gene is a frequent fusion partner in AML and ALL and shows two major breakpoint clusters⁸ of which we designed crRNAs for both (**Supplementary Table 1**). The B2 sample was randomly chosen out of a pool of six tumor samples (four ALL, one BL, one Burkitt’s-ALL) which could potentially harbor a BCR-ABL1, IGH/MYC or CRLF2-P2RY8 rearrangement. Therefore, we targeted the B1 sample with two crRNAs and the B2 sample with 14 crRNAs (**Supplementary Table 1**) and sequenced both samples on one flow cell each. NanoFG identified a KMT2A-MLLT6 fusion in B1 (**Figure 5A**) and a BCR-ABL1 fusion in B2 (**Figure 5B**) with 29 fusion-spanning and 27 fusion-spanning reads, respectively (**Figure 5C**). Overall, we observed a breakpoint-spanning enrichment of 938x (B1) and 313x (B2) and a fusion-spanning enrichment of 143x (B1) and 148x (B2) (**Figure 5D**). This demonstrates the capacity of FUDGE to identify unknown fusion events from tumor material.

Figure 5: Fusion gene coverage and enrichment from blinded samples and run-time analysis

(A) Coverage plots for the B1 tumor sample for the two fusion partners KMT2A (targeted) and MLLT6. (B) Coverage plots for the B2 tumor sample for the two fusion partners BCR (targeted) and ABL1. BCR was targeted with eleven sequential guides to span the possible breakpoint region. (C) Mean coverage and (D) mean enrichment across the genome, on-target (from cut to breakpoint), BP-spanning (breakpoint-spanning), and fusion-spanning

(across the fusion junction) for the tumor samples B1 and B2. (E) Time-course experiment on sequencing time necessary to identify fusion-spanning reads (%) and (F) fusion-spanning reads ($n = 2$). Dotted lines (green) indicate the crRNA-directed Cas9 cleavage positions and dashed lines (red) indicate breakpoint positions. Arrows indicate the directionality of reads created from the specific crRNA design.”

Results, page 16:

Fusion gene detection from low input tumor material

“Therefore, we sequenced WGA DNA of two colon cancer samples (C1 and C2), known to harbor BRAF fusions (AGAP3-BRAF and TRIM24-BRAF, respectively)⁹, a sarcoma sample with a SS18-SSX1 fusion (S1) and a PDX sarcoma sample with unknown fusion status (S2) We targeted the S2 sample with nine crRNAs targeting the most common recurrent sarcoma fusion partners EWSR1, PAX3, PAX7, and SS18. For all samples we performed WGA on 10 ng starting material and subjected 1 ug of WGA-DNA to the enrichment protocol. Genome coverage (**Figure 6A**) and read-length were comparable to previous experiments (**Supplementary Table 1**). Initially NanoFG did not detect the the AGAP3-BRAF fusion, however, lowering the threshold to one fusion-supporting read identified the fusion gene (**Figure 6A-C**). The TRIM24-BRAF fusion was called by NanoFG with 11 fusion-spanning reads (**Figure 6A-B, 6D**). For the S1 and the S2 sample, neither NanoFG nor manual inspection in IGV could detect a targeted fusion gene.”

2) As stated above, the assay appears most important in its ability to detect the DNA breakpoints for translocations occurring in solid tumors. Few sites are tested, and one (SS18-SSX), appeared to work poorly. Additional successful testing in these tumor types would be most demonstrative of the assay’s value.

Answer: The sample referred to by the reviewer showed overall little sequencing throughput (Supplementary Table 1) which could be the main reason for the poor performance of this specific translocation. Furthermore, the distance between cut and genomic breakpoint location is rather large (5885 kb), presumably lowering the sensitivity of the assay (see also point 2 reviewer 1). We have repeated the assay for this specific tumor type and fusion gene configuration by sequencing a sarcoma tumor sample with a SS18-SSX1 fusion gene. Since not enough starting material was available, we performed whole genome amplification on this sample. Unfortunately, after sequencing the SS18-SSX1 fusion could not be detected. The average read-length for this sequencing run was 1,9 kb, which is problematic to detect breakpoints further away from the cut-site. We do think that for SS18-SSX1 fusion, additional crRNAs need to be designed in order to obtain a robust assay. However, due to the lack of tumor material, we could not resequence this sample.

3) The assay appears to require fresh specimens, though this does not appear to be explicitly stated. This is a MAJOR limitation in the diagnostic context, particularly as it applies to solid tumor excisions and biopsy specimens, and would be the death knell for the technique if it merely identified the fusion genes, and not the precise genomic

breakpoints, as most assays do. The authors need to highlight this in their discussion.

Answer: We agree with the reviewer and have now stated the requirement for fresh specimens specifically in the text (see also point 4 reviewer 1).

Textual changes

Results, page 5:

*“To achieve this, genomic DNA isolated from fresh frozen samples is dephosphorylated as previously described⁵ and a crRNA flanking the suspected breakpoint region(s) is utilized to target Cas9 to a specific genomic loci where it creates a double-strand DNA break (**Figure 1A**).”*

Discussion, page 21:

“A limitation of this approach is the requirement of non-fragmented DNA. Applying the FUDGE crRNA protocol to FFPE material (the current standard for pathology procedures), will most likely fail to comprehensively identify fusion genes due to short read lengths derived from degraded FFPE DNA. An adaptation of the design strategy to regularly interspace crRNAs at short intervals may overcome this issue; however, this approach will drastically increase the assay costs per fusion gene.”

4) There is no justification given for the two read cutoff for calling fusion genes. Were fusion negative specimens tested to arrive at this number? Is this sufficiently specific? It's very difficult to evaluate specificity from the current presentation of the data, which currently focuses only on sensitivity. Two reads is an unusually low threshold for a sequencing related test, and may be unreasonable in the setting of WGA.

Answer: We have included an experiment giving rationale for the two read cut-off (Supplementary Figure 5D). We chose two unamplified and two WGA-samples and reran NanoFG with fusion-supporting read thresholds from $n = 1$ to $n = 5$. It is apparent that one fusion-supporting read leads to a high number of false-positives (as these involve genes not targeted in the assays). In the range between two and five fusion-supporting reads, the number of detected fusions decreases substantially as compared to one fusion-supporting read, while the overall number of detected fusions within this range stays stable. Hence, we provide a rationale that a two read cut-off offers the optimal high true-positive and low false-positive results.

Textual changes

Results, page 16:

*“Notably, WGA introduced accompanying structural variation leading to a high number of fusion gene predictions (**Supplementary Figure 5D**) and difficulties for manual inspection in IGV. However, we show that a fusion supporting threshold of two reads is a reasonable cut-off for normal and WGA-samples, as the number of predicted fusions decreases drastically compared to one supporting read but remains relatively stable compared to a higher fusion-*

support (**Supplementary Figure 5D**). Furthermore, fusion genes identified by NanoFG that were not targeted through crRNAs within our assay are very likely to be false-positives.”

Supplemental Figure 5D Graph depicting the number of fusion genes called by NanoFG depending on the set threshold of fusion-supporting reads ($n=1-5$) for the samples CHP-100, ES1, C1-WGA and C2-WGA.

5) A more clinically applicable implementation of multiplexing, if performed, would be helpful. If not the discussion regarding the ease of deconvolution of pooled specimens is, I think, overstated. Ideally testing should be performed on patient specimens, not cell lines.

Answer: We agree that the current multiplexing procedure may be suboptimal for clinical purposes. Unfortunately, barcoding techniques offered by Oxford Nanopore Technologies (ONT) are currently not compatible with this protocol. ONT is/will be releasing the Flongle (a cheap and smaller flow cell, which produces less throughput) which would be the optimal solution to test each clinical sample individually without the need to multiplex for cost-saving purposes. We have rewritten the discussion, highlighting the drawbacks of the current multiplexing approach but explaining future possibilities.

Textual changes

Discussion, page 21:

“Our current multiplexing approach, with sample pooling and retrospective demultiplexing by breakpoint PCR, reduces cost but prolongs assay duration and increases the complexity of sample processing. With lower throughput flow cells, such as the ONT Flongle, individual samples could be run separately, without pooling and demultiplex PCR, thus simplifying the workflow and lowering assay costs dramatically.”

6) Finally, there are two areas (mentioned above) where this assay has immense promise, but that promise is not explored. These are in MRD detection and detection of non-fusion-producing translocations. Any material exploration of these would substantiate the authors’ claims of clinical utility.

Answer: We thank the reviewer for this comment. We agree that the detection of genomic breakpoints with this assay has immense potential for diagnostic applications such as MRD tracing. In a study, which is currently being evaluated, we show the feasibility of utilizing the genomic breakpoint junctions detected by nanopore sequencing as biomarkers for MRD

tracing in liquid biopsies³. As mentioned in Point 4 by reviewer 1, we have now highlighted the rationale for genomic breakpoint detection in the main text.

Textual changes

Discussion, page 20:

“Rapid identification of the genomic breakpoint offers the opportunity to utilize the breakpoint junctions as a biomarker for minimal residual disease (MRD) tracing³. Common diagnostic approaches for fusion gene detection can be divided into DNA or RNA-based approaches (Table 1). Detection of fusion genes on the RNA level might be less complex due to the restriction of breakpoints to exon-exon junctions; however, RNA molecules are less stable and the overall abundance is influenced by expression levels. DNA-based approaches such as targeted NGS assays or WGS are preferable since they identify all fusion gene events including promoter fusions as well as the exact genomic breakpoint. However, these assays are hampered by longer turn-around times and WGS can result in high false-positive rates.”

Table 1

	DNA				RNA		
Stability	Stable				Unstable		
Breakpoint	Across large genomic areas				Only exon-exon junctions		
Detection method	FISH	FUDGE	Targeted-NGS	WGS	RT-PCR	Targeted-NGS	RNA seq
Speed (days)	1	2	7-14	7-14	1	7-14	7-14
Targeted	Semi	Semi	Semi	No	Yes	Semi	No
Partner detection	No	Yes	Yes	Yes	N/A	Yes	Yes
Promoter fusions	No	Yes	Yes	Yes	No	No	No
Validation required	N/A	No	No	Yes (Sanger)	No	No	Yes (Sanger)
MRD	No	Yes	Yes	Yes	Yes	Suboptimal	Suboptimal

Discussion, page 21-22:

“The application of this assay in the clinic could allow for rapid gene fusion detection to allow appropriate therapy initiation and identification of specific genetic targets for blood-based minimal residual disease tracing.”

Minor concerns:

1) The format of the flow cell should be mentioned in the main body of the paper, as there is over an order of magnitude range of costs for a “nanopore flow cell” at this time. It should be clear that this describes work with the R9.4 and not flongle flow cell (which would be more impressive and more clinically plausible). This has major cost implications.

Answer: We have incorporated the version of the flow cell in the main text. As mentioned in point 5 of the major concerns, we agree that using a flongle flow cell would be optimal, however, thus far availability of these flow cells has been limited due to high demand and we have not been able to obtain flongles for evaluation.

Textual changes

Introduction, page 3:

“However, sequencing throughput from one Nanopore flow cell (2-5x genome coverage; R9.4) is insufficient to elucidate the complete structural variation (SV) landscape of a genome¹⁰.”

Results, page 5:

“Thereafter, the enriched libraries are sequenced on one ONT flow cell (R9.4).”

2) Some brief examination of the run time of NanoFG pipeline should be described. Apologies if I have missed it.

Answer: The run time of NanoFG is described on the GitHub page¹¹ but is now also mentioned in the material and methods section of the manuscript.

Textual changes

Material and Methods, page 27:

“The run time of NanoFG on +-25,000 nanopore reads is approximately 20 minutes using a single thread. Detailed instructions including a test-set can be found on GitHub (<https://github.com/SdeBlank/NanoFG>).”

3) The authors need to make it clear in the introduction and discussion that this does not comprise the only method for targeted analysis of fusion breakpoints using high throughput sequencing. Many of these methods similarly have a fairly rapid turnaround time, show a higher degree of target enrichment, have a broader target

panel, and have been demonstrated and are regularly used in a clinical setting. To the uninformed, it currently reads as if such options do not exist.

Answer: We thank the reviewer for this comment and the similar comment of Reviewer 1 (point 3), and agree that highlighting current methods and their similarities/differences to FUDGE is necessary to put FUDGE in context with current practices. To address this, we have modified the text and describe other targeted fusion gene detection NGS approaches. Finally, we discuss the advantages and disadvantages of these different techniques with respect to FUDGE (see also point 4 reviewer 1).

Textual changes:

Introduction, page 3:

“Recently, next generation sequencing (NGS) assays which specifically target recurrent fusion partners have been developed and are currently implemented in clinical practice^{1,2}. These assays are highly versatile with respect to partner identification and input material (e.g. suitable for DNA isolated from Formalin-Fixed Paraffin Embedded tissue blocks; FFPE), but are accompanied with longer turnaround-times, increased costs and bioinformatic challenges.”

Discussion, page 20:

*“Common diagnostic approaches for fusion gene detection can be divided into DNA or RNA-based approaches (**Table 1**). Detection of fusion genes on the RNA level might be less complex due to the restriction of breakpoints to exon-exon junctions; however, RNA molecules are less stable and the overall abundance is influenced by expression levels. DNA-based approaches such as targeted NGS assays or WGS are preferable since they identify all fusion gene events including promoter fusions as well as the exact genomic breakpoint. However, these assays are hampered by longer turn-around times and WGS can result in high false-positive rates.”*

4) The paper mentions that false positives were identified when performing WGA prior to NanoFG. Additional data regarding the rate of these false positives should be given and it should be stated whether or not any false positives occurred in the target area (it appears that they were not, but it’s not explicitly stated).

Answer: In line with the explanation for the incentive to use a two read cut-off (point 4, reviewer 2, major comments), we have provided an overview of how many false positives were reported without and upon WGA (Supplemental Figure 5D).

Textual changes

Results, page 16:

*“Notably, WGA introduced accompanying structural variation leading to an high number of fusion gene predictions (**Supplementary Figure 5D**) and difficulties for manual inspection in IGV. However, we show that a fusion supporting threshold of two reads is a reasonable cut-off for normal and WGA-samples, as the number of predicted fusions decreases drastically compared to one supporting read but remains relatively stable compared to a higher fusion-*

support (**Supplementary Figure 5D**). Furthermore, fusion genes identified by NanoFG that were not targeted through crRNAs within our assay are very likely to be false-positives.”

Supplemental Figure 5D Graph depicting the number of fusion genes called by NanoFG depending on the set threshold of fusion-supporting reads (n=1-5) for the samples CHP-100, ES1, C1-WGA and C2-WGA.

Additional changes:

In addition to the responses to the reviewer comments, we have made three changes to the current version of the manuscript that we believe add to the value of the manuscript.

- 1) *As suggested by Reviewer 1 we now report the breakpoint-crossing reads in addition to the fusion-spanning reads. In the previous version of the manuscript, we also reported the target-locus (+/- 10 kb to either side of the cut) coverage and enrichment. We have removed this parameter because it does not offer additional relevant information and may lead to confusion for the reader.*
- 2) *We changed the parameters of NanoFG, more specifically within NanoSV to increase the cluster distance of breakpoint-junctions detected per fusion from 10 to 100 bp. We observed that sometimes, presumably due to inaccurate mapping, reads that support the same fusion breakpoint do not fall within this 10 bp region. Enlarging this area identifies, in some cases, more fusion gene supporting reads that support the same breakpoint. Therefore, some numbers from the old version to new version of the manuscript have been adapted.*
- 3) *In this version of the manuscript we include an agarose gel picture validating the SS18-SSX1 fusion gene (HS-SYII cell line), which was detected by FUDGE with only one fusion supporting read (**Supplementary Figure 3B**).*
- 4) *In the previous version of the manuscript, the Monomac-1 cell line was positive for the MLLT3-KMT2A, KMT2A-MLLT3 as well as the KMT2A-MLLT1 fusion gene (**Supplementary Figure 7A**). The latter was a contamination from the KOPN8 cell line. We have isolated fresh DNA from the Monomac-1 cell line, performed the breakpoint PCRs again and show that this cell line is -as expected¹²- only positive for the MLLT3-KMT2A and KMT2A-MLLT3 fusion genes (**Supplementary Figure 7B**).*

References

1. Lam, S. W. *et al.* Molecular Analysis of Gene Fusions in Bone and Soft Tissue Tumors by Anchored Multiplex PCR-Based Targeted Next-Generation Sequencing. *J. Mol. Diagn.* **20**, 653–663 (2018).
2. Wang, K. *et al.* Patient-derived xenotransplants can recapitulate the genetic driver landscape of acute leukemias. *Leukemia* **31**, 151–158 (2017).
3. Valle-Inclan, J. E. *et al.* Rapid identification of genomic structural variations with nanopore sequencing enables blood-based cancer monitoring. *medRxiv* 19011932 (2019).
4. Olsson, E. *et al.* Serial monitoring of circulating tumor DNA in patients with primary breast cancer for detection of occult metastatic disease. *EMBO Mol. Med.* **7**, 1034–1047 (2015).
5. Gilpatrick, T. *et al.* Targeted nanopore sequencing with Cas9-guided adapter ligation. *Nat. Biotechnol.* **38**, 433–438 (2020).
6. QIAGEN. Whole Genome Amplification — Overview - QIAGEN. <https://www.qiagen.com/nl/service-and-support/learning-hub/technologies-and-research-topics/wga/overview-on-wga/>.
7. Marshall, A. D. & Grosveld, G. C. Alveolar rhabdomyosarcoma – The molecular drivers of PAX3/7-FOXO1-induced tumorigenesis. *Skeletal Muscle* vol. 2 25 (2012).
8. Meyer, C. *et al.* Human MLL/KMT2A gene exhibits a second breakpoint cluster region for recurrent MLL-USP2 fusions. *Leukemia* (2019) doi:10.1038/s41375-019-0451-7.
9. Kloosterman, W. P. *et al.* A Systematic Analysis of Oncogenic Gene Fusions in Primary Colon Cancer. *Cancer Res.* **77**, 3814–3822 (2017).
10. Jain, M. *et al.* Nanopore sequencing and assembly of a human genome with ultra-long reads. *Nature Biotechnology* vol. 36 338–345 (2018).
11. SdeBlank. SdeBlank/NanoFG. *GitHub* <https://github.com/SdeBlank/NanoFG>.
12. German Collection of Microorganisms and Cell Cultures GmbH: Details. <https://www.dsmz.de/collection/catalogue/details/culture/ACC-252>.

Reviewers' Comments:

Reviewer #1:

Remarks to the Author:

Dear Authors,

Thank you for addressing my comments sufficiently, and for including the additional analysis and experiments.

I look forward to using your approach within our laboratory shortly.

Reviewer #2:

Remarks to the Author:

In this revised manuscript "Partner independent fusion gene detection by multiplexed CRISPR Cas9 enrichment and long read Nanopore sequencing", the authors provide additional data, further experimental results, and textual changes to support their new, CRISPR-Cas9 based method for the rapid detection of fusion events.

The authors have addressed my previous concerns. I think this is an excellent candidate for publication, and is an important result that will be quite interesting to the field. I hope that it will be accepted for publication.

I offer only my following comments, none of which preclude publication at this time.

1) While the authors have expanded their testing of multiple mutations, the "blinding" of the sort that they employed is not quite what I'd intended (partly due to my lack of clarity). In a clinical diagnostic context, often the most important thing is ascertaining the presence or absence of a particular fusion or confined set of fusions, which is often not known from the outset, or not known with certainty. Testing with a larger number of positive and negative samples with unknown diagnosis and origin (to those doing the analysis) would have been more convincing. However, I am impressed by the additional data. Further validation is out of the scope of the current publication, and I look forward to further developments and validation of this method in a scenario that more closely approximates the clinical context.

2) I now better understand the issues with the specimen with SS18-SSX translocation, and do not think that this precludes publication.

3) The details on the limitations of the assay, e.g. need for fresh tissue, are now better handled in the text. I appreciate these additions.

4) The supplementary material is helpful to explain the thresholding choices for fusion calling. The specificity at 2 reads is impressive. I suppose the lack of a PCR step aids in this. This is a nice advantage of the assay.

5) The multiplexing issues are understandable. The flongle is really the true solution to the cost issue. Future work reporting on implementation in that platform would be welcome.

6) In my judgement the cost of this assay and limitations regarding source material will NOT make it a widely adopted tool in the diagnostic context. The additional text regarding MRD detection is also welcome. The additional testing in a fusion that would NOT produce a fusion gene, and therefore not be detectable by RNA based tools, is also a good addition.

All minor concerns are addressed.